# Inferring control objectives in a virtual balancing task in humans and monkeys

Mohsen Sadeghi[1†], Reza Sharif Razavian[1,2,3†], Salah Bazzi[1,2,4*],
Raeed H Chowdhury[5], Aaron P Batista[5], Patrick J Loughlin[5], Dagmar Sternad[1,2,4,6]

[1]Department of Biology, Northeastern University, Boston, United States; [2]Department of Electrical and Computer Engineering, Northeastern University, Boston, United States; [3]Department of Mechanical Engineering, Northern Arizona University, Flagstaff, United States; [4]Institute for Experiential Robotics, Northeastern University, Boston, United States; [5]Department of Bioengineering, and Center for the Neural Basis of Cognition, University of Pittsburgh, Pittsburgh, United States; [6]Department of Physics, Northeastern University, Boston, United States

**Abstract** Natural behaviors have redundancy, which implies that humans and animals can achieve their goals with different strategies. Given only observations of behavior, is it possible to infer the control objective that the subject is employing? This challenge is particularly acute in animal behavior because we cannot ask or instruct the subject to use a particular strategy. This study presents a three-pronged approach to infer an animal's control objective from behavior. First, both humans and monkeys performed a virtual balancing task for which different control strategies could be utilized. Under matched experimental conditions, corresponding behaviors were observed in humans and monkeys. Second, a generative model was developed that represented two main control objectives to achieve the task goal. Model simulations were used to identify aspects of behavior that could distinguish which control objective was being used. Third, these behavioral signatures allowed us to infer the control objective used by human subjects who had been instructed to use one control objective or the other. Based on this validation, we could then infer objectives from animal subjects. Being able to positively identify a subject's control objective from observed behavior can provide a powerful tool to neurophysiologists as they seek the neural mechanisms of sensorimotor coordination.

*For correspondence:
s.bazzi@northeastern.edu

†These authors contributed
equally to this work

Reviewing Editor: Noah
J Cowan, Johns Hopkins
University, United States

## eLife assessment

This study represents a step towards integrating human and non-human primate research towards a broader understanding of the neural control of motor strategies. It could offer **valuable** insights into how humans and non-human primates (Rhesus monkeys) manage visuomotor tasks, such as stabilizing an unstable virtual system, potentially leading to discoveries in neural behaviour mechanisms. While the evidence is mostly **solid**, some results, particularly from the binary classification of control strategies for non instructed behaviour, require further validation before it could be conclusively interpreted.

## Introduction

Almost all actions in daily life can be achieved in multiple ways that all can lead to the desired task goal. As an example, consider a driver steering a car on a curvy road to reach a target destination. She may choose different paths depending on whether she wants to maintain a consistent distance from the median strip or whether she aims to minimize changes in velocity. Both strategies can achieve her

goal, i.e., arrive at her destination, maybe even arriving at the same time, although the precise path taken by the car in both situations will differ. How could one identify the underlying control objective from differences in observed behavior? In more technical terms, what is the objective/cost function that an individual aims to achieve/minimize to accomplish a task? A considerable number of studies in human movement neuroscience have aimed to identify the control objectives in a given task based on their kinematic manifestations (*Braun et al., 2009*; *Izawa et al., 2008*; *Nagengast et al., 2009*; *Razavian et al., 2023*; *Uno et al., 1989*; *Wong et al., 2021*). However, experimental tasks are often chosen to elicit consistent behavioral features across repetitions and individuals, not only to facilitate analysis, but also to constrain control to a single objective. Behavior in natural settings, however, tends to be more complex and highly variable across repetitions, because there is redundancy, meaning that a variety of ways exist to achieve the goal. Hence, individuals can employ a multitude of strategies to accomplish a task. To date, the understanding of such variable behaviors with underlying redundancy - let alone its neural bases - has posed formidable challenges (*Croxson et al., 2009*; *Diedrichsen et al., 2010*; *Kawato, 1999*; *Scott, 2004*).

Attempts to understand the neural underpinnings of control objectives have been pursued in research on both humans and non-human primates (*Benyamini and Zacksenhouse, 2015*; *Cross et al., 2023*; *Croxson et al., 2009*; *Desrochers et al., 2015*; *Kao et al., 2021*; *Miall et al., 2007*; *Nashed et al., 2014*; *Omrani et al., 2016*). Yet, with notable exceptions (e.g. *Pruszynski et al., 2011*), these two lines of inquiry have remained largely parallel with few direct bridges: human behavioral and computational research has mainly focused on the analysis of behavior, while animal research has used invasive methods such as intracortical recordings to understand the neural mechanisms of movement control. Experiments with humans tend to use detailed experimental manipulations to elicit features of motor behavior that afford insights into its governing principles. Using a wide range of tasks, from simple reaching to interacting with complex objects, mathematical models with specific control algorithms have been used to reproduce the salient features of behavior (*Crevecoeur et al., 2019*; *Diedrichsen, 2007*; *Nagengast et al., 2009*; *Nayeem et al., 2021*; *Razavian et al., 2023*; *Yeo et al., 2016*). However, understanding the neural underpinnings of movement control at the intracortical level in healthy humans has remained a challenge. On the other hand, animal research, in particular with non-human primates, allows sophisticated methods to directly record neural activity to afford insights into neural correlates of motor behavior. Ultimately, this knowledge should transfer to how the human brain functions (*Badre et al., 2015*), but those links have only been made few and far between.

To achieve this goal, cooperative study designs between human and animal motor research are needed to understand the neural basis of human motor skill (*Badre et al., 2015*; *Rajalingham et al., 2022*). However, there are difficult challenges to overcome: First, cooperative design requires matching behavioral tasks that can be performed similarly and with the same conditions by both humans and animals. While research on eye movement control has achieved such matching between human and non-human primate paradigms (e.g. the anti-saccade task or the Rashbass step ramp; *Lisberger et al., 1987*; *Munoz and Everling, 2004*; *Rashbass, 1961*; *Robinson, 2022*), this proves more challenging in limb coordination, where explicit goals and instruction become more important. Second, the constraints of behavioral studies with monkeys and humans are somewhat different, which can preclude a direct comparison. Behavioral tasks used with monkeys are typically simpler than those used with humans, due to the animals' more limited cognitive capacities. Also, studies with monkeys aim for highly repeatable behaviors to facilitate the aggregation of neural activity across trials or days. This means that tasks with redundancy that allow multiple solutions to achieve the same goal do not readily lend themselves to investigation. In contrast, studies of human behavior can push toward tasks that are more sophisticated and capture the complexity and redundancy that abound in natural activities. Our experiments examine a behavioral task with redundancy that allows more than one solution to accomplish the task. Despite these challenges, our study aims to bridge the gap between human and monkey behavioral studies to build toward an understanding of the neural principles of human motor control.

We used an experimental paradigm, the Critical Stability Task (CST), that can be performed by both humans and monkeys (*Quick et al., 2018*). The CST requires the subject to balance an unstable virtual system governed by a very simple dynamical equation (see Methods). Performing the task is akin to balancing a virtual pole. The CST has features that make it suitable for the study of more complex

motor behaviors. First, while the goal remains the same, the difficulty of the task can be titrated. Second, it involves interactions with an object, albeit virtual in our case, so that continuous adjustments are required to succeed. Each trial evokes unique behavior that may reflect different control strategies to accomplish the task. In addition, even if the same control strategy is employed, each trial generates different behavior due to sensorimotor noise and the task's instability. These features are ubiquitous in all everyday actions and our choice of CST was to explicitly address such behavior that is closer to naturalistic behaviors. As in the car driving analogy, the subjects might seek to optimize position, or they might seek to optimize velocity, while both strategies may lead to equal success.

Because of its complexity and redundancy, each trial of the CST is unique. The goal of the study is to infer the subject's control objective (i.e. minimization of errors in position or velocity) from observations of their behavior. When the subjects are humans, it is possible to instruct them to employ a particular strategy or to ask them post-hoc what strategy they adopted to succeed at the task. This explicit route is definitely not available with monkeys. As we are still quite far from 'reading out' strategies from neural activity, we need to start with behavior to infer the control objectives. Hence, this study adopted a computational approach based on optimal control theory to simulate behavior during the CST in various conditions. This approach allowed us to make predictions about the behavioral signatures associated with different control policies, which we then used to analyze the experimental data from both humans and monkeys.

In overview, this study investigated, through behavioral data and model-based simulations, the sensorimotor origins of observed kinematic strategies in humans and non-human primates performing the CST. We developed the experimental paradigm such that humans and monkeys executed the task

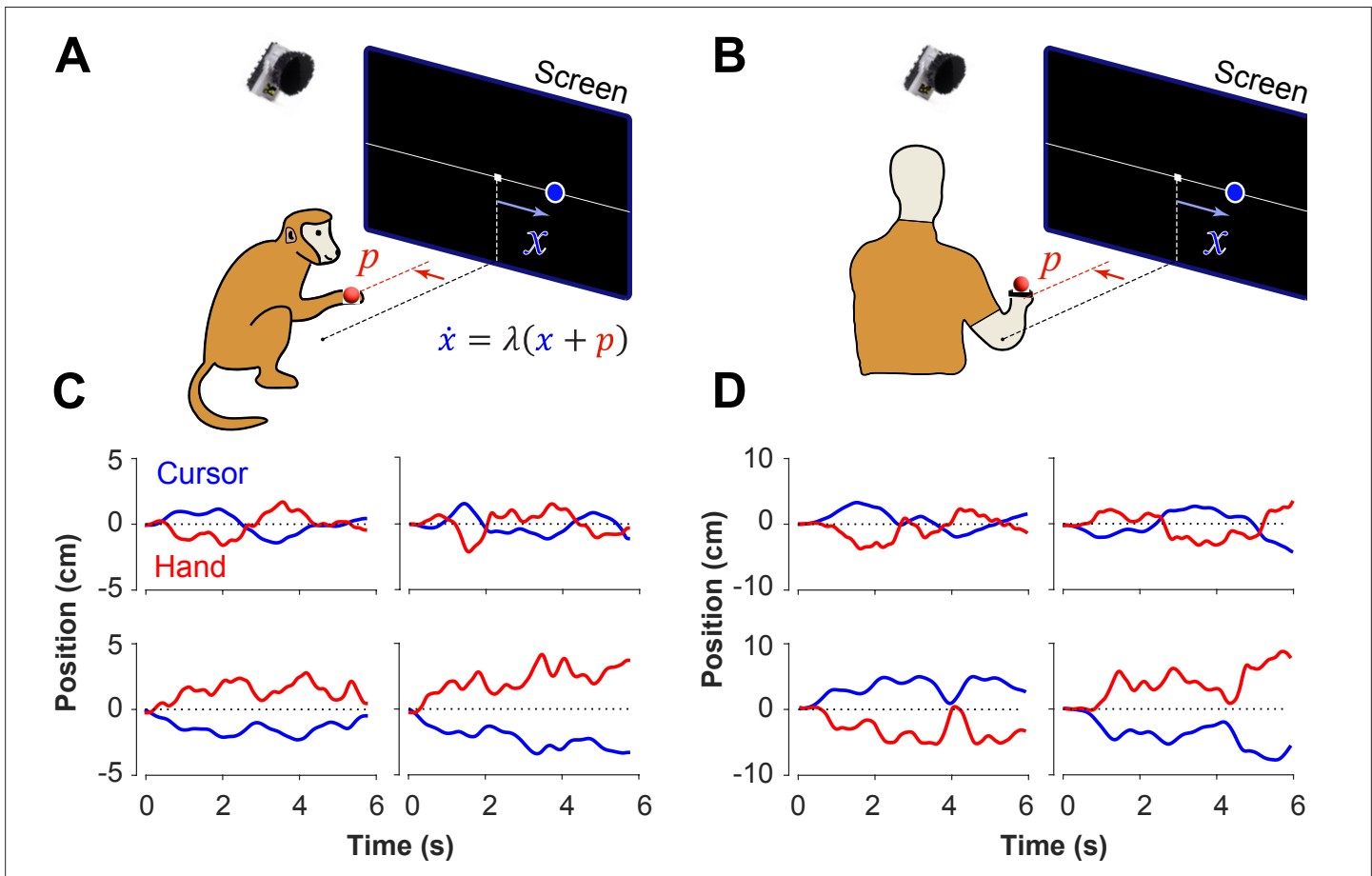

**Figure 1.** Experimental setup for monkeys and humans performing the CST. Monkeys (**A**) and humans (**B**) controlled an unstable cursor displayed on a screen using lateral movements of their right hand. The hand movements were recorded using motion capture; the data were used in real-time to solve for the cursor position and velocity through the CST dynamics equation. Timeseries of the hand (red) and cursor (blue) movements shown for four example trials from monkeys (**C**) and humans (**D**).

under matching conditions while recording movement kinematics in exactly the same way. An optimal control model was used to simulate two different control objectives, through which we identified these different objectives in the experimental data of humans and monkeys. We discuss how in the future these results could guide the analysis of neural data collected from monkeys to understand the neural underpinnings of different control policies in an interactive feedback-driven task with redundancy.

## Results

The CST involved balancing an unstable system using horizontal movements of the hand to keep a cursor from moving off the screen (*Figure 1A and B*). This study collected data from human subjects performing the CST and compared it to previously collected data from monkeys performing the same task. The hand's displacements were recorded by 3D motion capture (Qualisys, Gothenburg), with a reflective marker attached to the hand. The cursor dynamics were generated by a linear first-order dynamical system, relating hand and cursor kinematics as described in *Quick et al., 2018*:

$$\dot{x} = \lambda \left( x + p \right) \tag{1}$$

where $x$ and $\dot{x}$ are the horizontal cursor position and cursor velocity on the screen, $p$ is the horizontal hand position, and $\lambda$ is a positive constant fixed at the beginning of each trial. The parameter $\lambda$ sets the gain of the system. When $\lambda$ is larger, the cursor would tend to move faster, making the task more difficult as faster and more precise hand movements were required to maintain balance. Correspondingly, success rates at the task decreased with increasing $\lambda$. To summarize the skill of human and monkey participants, we identified the value at which subjects succeeded at only 50% of the trials and defined that value as the 'critical' value, $\lambda_c$.

The task goal was to keep the cursor within a range of space shown on the screen for a duration of 6 s. The range of the workspace was defined as $-c \leq x\left(t\right) \leq c$, where $c$ was a positive constant (c=5 cm or 10 cm; see Methods). This created a redundancy in achieving the task goal as there were infinitely many ways in which one could balance the cursor inside the specified region. We examined movement kinematics to identify control strategies employed by different subjects, and across different trials.

In a previous study, two Rhesus monkeys were trained to perform the CST under increasing difficulty levels (*Quick et al., 2018*). Similarly, here 18 human subjects were recruited to perform the same task under comparable experimental conditions as the monkeys (see Methods). *Figure 1* illustrates the experimental setup for both monkeys and humans (*Figure 1A and B*) and shows examples of their behavior (*Figure 1C and D*). Overall, there were similarities in performance between humans and monkeys. To further quantify and compare this performance across humans and monkeys, we defined a set of control metrics to assess different aspects of control as detailed in the following.

### Experiment 1: CST performance without instructed strategy

In the first experiment, six human subjects performed the CST with the only instruction to 'perform the task without failing to the best of your ability'. Failure occurred if the cursor escaped the boundaries of the screen (±10 cm from the center) within the trial duration of 6 s. Subjects received categorical feedback about the outcome at the end of each trial in a text appearing on the screen reading 'Well done!' for success, and 'Failed!' for failure. The degree of difficulty, set by $\lambda$, was increased stepwise across trials until the subject could no longer perform the task (see Methods for the specifics about the setting of $\lambda$ values).

We first sought to examine the main characteristics of behavior in CST performance and how it compared between humans and monkeys. To quantify the overall behavior, four main metrics were employed as described and motivated below. To begin, we considered the overall success rate in the task among different individuals, before focusing on the kinematics of task performance. *Figure 2A* illustrates the success rates and how they dropped as the task difficulty increased. Both humans and monkeys showed a similar pattern of decrease in success rate which was well-captured with a sigmoidal function. Expectedly, individuals varied in their ability to achieve high difficulty levels as a measure of skillful performance, indicated by their 'critical $\lambda$ value' $\lambda_c$, that is, the value of $\lambda$ when the success rate dropped below 50%.

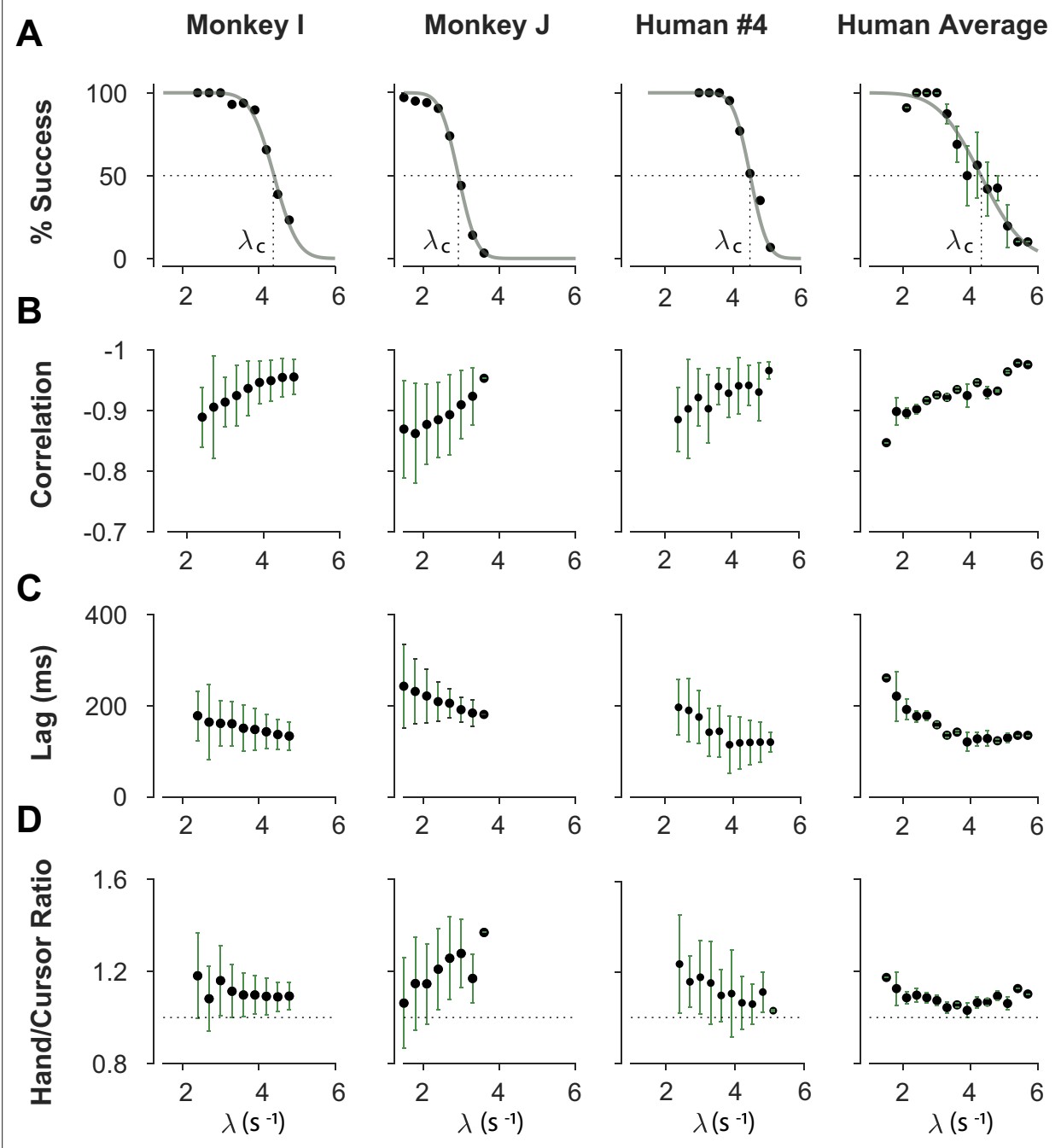

**Figure 2.** Overall behavioral characteristics of CST performance as a function of task difficulty (λ). Data is shown for two individual monkeys (first two columns from left) from a previous study (*Quick et al., 2018*), as well as an example human individual (third column from left) and the average across human subjects (right-most column; n=6). For the individual subjects, each data point and its corresponding error bars represent the mean ± SD across trials for any given difficulty level, respectively. For the human average plot, the data points and their corresponding error bars represent the mean ± SE across individuals for each difficulty level. (**A**) Psychometric curves for success rate (%) as a function of task difficulty (λ). The difficulty level at which the success rate crossed 50% was considered as the critical stability point ($λ_c$), indicating the individual's skill level in task. (**B**) Correlation between the hand and cursor position trajectories during CST. (**C**) Sensorimotor lag between the cursor and the hand movements. (**D**) Ratio of hand RMS over the cursor RMS calculated for each trial, representing the strength of the hand response relative to the cursor displacement.

To investigate the performance in more detail, the kinematics of movement were examined, specifically the hand and cursor position during each trial. As indicated in *Equation 1*, the hand position $p$ was the control input to the system which aimed to control the cursor position $x$ as the variable of interest. Due to the unstable nature of the task, drifting of the cursor towards the edge of the screen

demanded a response by the hand movement to avoid failure. As such, two simple metrics characterized control, one quantifying how the movement of hand and cursor correlated, and a second one to what degree the hand response lagged cursor displacements. *Figure 2B* shows the correlation between the cursor and hand movements as a function of task difficulty. The strength of the correlation increased as trials became more challenging in both monkeys and humans, asymptoting towards –1. According to *Equation 1*, this behavior was equivalent to reducing the sum $(p + x)$ to mitigate the effect of large $\lambda$ values on cursor velocity $\dot{x}$ , and, hence, reduce the chance of failure.

The response lag from the cursor movement (observed feedback) to the hand movement (control response) is an important characteristic of a control system. As shown in *Figure 2C*, by increasing the task difficulty $\lambda$, the lag decreased for all subjects. Coupled with the increase in the strength of the correlation with increasing $\lambda$, these findings indicate that subjects generated faster and more precise corrective responses to cursor displacements in more difficult trials.

As the fourth metric, we calculated the ratio of root mean squared (RMS) of hand position to the RMS of cursor position for each trial, as a measure of response strength. This measure determined to what extent the control signal (hand movement) compared in magnitude to the cursor movement. A large RMS ratio meant that on average across a trial, the hand exhibited larger movements than necessary to correct for cursor deviations. *Figure 2D* illustrates the calculated RMS ratio as a function of task difficulty for humans and monkeys. Except for *Monkey J*, the RMS ratio showed a gradual decrease as the task difficulty increased for most individuals. The seemingly divergent behavior of *Monkey J* was likely due to subject-to-subject variability, as also observed in human performers (*Appendix 1—figure 1*). Such decrease could be justifiable due to larger cursor movements at higher difficulty levels, and perhaps more *efficient* corrective hand responses to cursor displacements. It is worth noting that for high $\lambda$ values, small hand movements could cause large cursor displacements, which were detrimental to the task success. Therefore, pruning any task-irrelevant hand movements, consistent with promoting efficiency, seemed essential to succeed in more difficult trials.

Overall, the control metrics presented in *Figure 2* give insight into how the CST was performed: as the task difficulty increased, subjects tended to respond to cursor displacements faster (that is, with lower lag), more precisely (seen in the stronger hand-cursor correlation), and more efficiently (with lower RMS ratio). Behavior was comparable between humans and monkeys, which suggests that there were similar control strategies used by both species. Next, we sought to detect those control strategies.

## Redundancy of control strategies in CST performance

The CST, as described earlier, affords redundancy in the behavioral strategies that could result in task success. Although covert in aggregate level of performance (i.e. *Figure 2*), single trial observations of hand and cursor trajectories suggested that different underlying control objectives might be at play. Two types of behavioral patterns appeared recognizable in the data. In one case, the cursor seemed to be always balanced around the center of the screen, and any deviations from the center induced a response to bring the cursor back to the center. This was reflected in the oscillatory movements of the cursor around the center, shown in example trials in *Figure 1C and D* (first row). In other trials, the cursor either exhibited a slow drift from the center or remained relatively still anywhere within the boundaries of the screen, with only limited attempts to bring the cursor back to the center (for example, *Figure 1C and D*, second row). We hypothesized that these patterns of behavior arise from different control objectives, each focused on a different state variable in the state-space of the cursor movement. In the former case, the position of the cursor appeared to be the primary control variable. Under this strategy, subjects might pursue the objective of keeping the cursor near the center of the screen. We refer to this strategy as Position Control. In the latter case, the cursor velocity seemed to be of primary importance for control, with the objective to slow down cursor velocity regardless of its position in the workspace. We refer to this strategy as Velocity Control.

Can we distinguish between different control strategies by examining behavior? To test this idea, we took a computational approach by developing a generative model based on optimal feedback control (*Todorov and Jordan, 2002*) that could simulate the task under different conditions and with different objectives (*Todorov and Jordan, 2002*). The model involved a controller that generated optimal motor commands based on a given objective to perform the CST via a simple effector model. The model also contained a state estimation block that estimated the states of the system based on

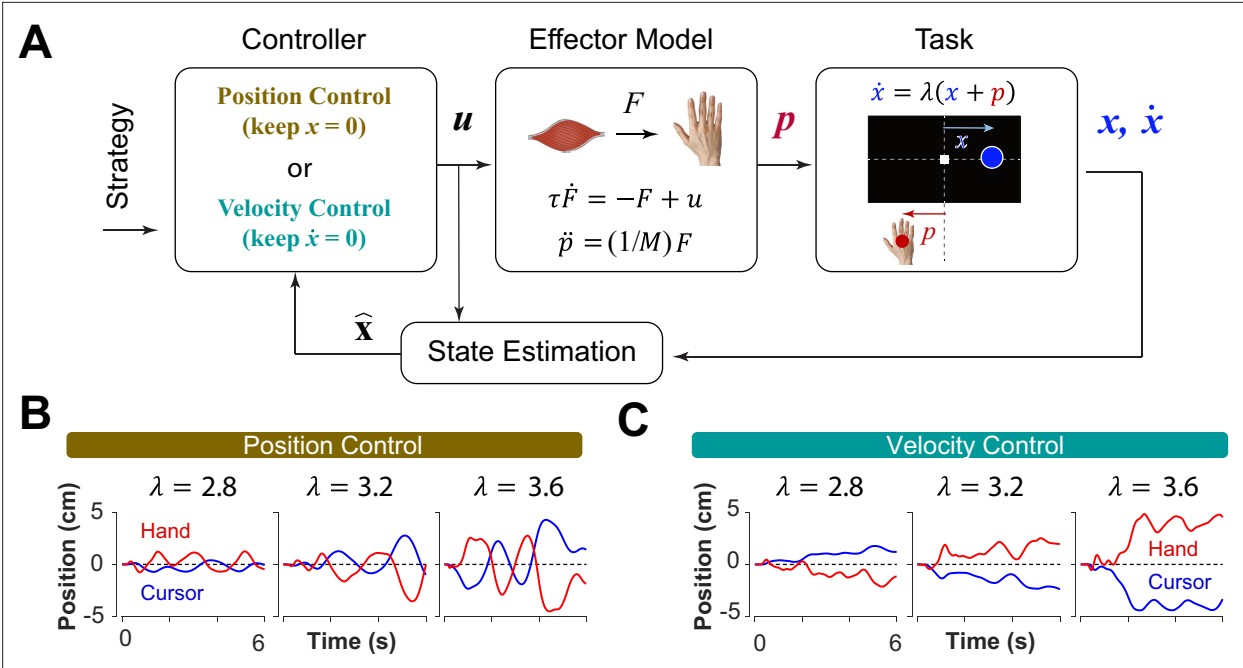

**Figure 3.** A generative model to perform the CST. (**A**) An optimal feedback controller generates motor commands based on two control objectives, position and velocity control. The motor command leads the movement of the effector (hand), which performs the CST. The cursor position and velocity are provided as feedback from which all the states are estimated and fed back to the controller. (**B, C**) Example trials simulated under the two control objectives for different difficulty levels: keeping the cursor at the center (**B**; Position Control) and keeping the cursor still (**C**; Velocity Control).

the given feedback (**Todorov, 2005**). In this case, cursor position and cursor velocity were used as feedback to the controller at each time step. **Figure 3A** illustrates a block diagram of this model.

The control gains used in the controller to generate the motor commands were optimally found by minimizing the sum of two cost functions: the cost of effort to reduce energy, as well as the cost of accuracy that prevented the states of the system from making large deviations (2):

$$J = \sum_{t=1}^{n} \left( \mathbf{x}_t^T Q \mathbf{x}_t + u_t^T U u_t \right) \tag{2}$$

where $u$ and $x$ represented the motor command and the state vector of the system, respectively. In this model, the state vector consisted of six states: the position, velocity and acceleration of the hand, as well as the position, velocity and acceleration of the cursor (see Methods). Variables $t$ and $n$ represented the time, and the total number of time steps in a trial, respectively. The matrix $Q$ and the scalar $U$ determined the weight of accuracy and effort in the cost function, respectively. Importantly, the matrix $Q$ allowed for determining which states of the system were of primary importance in the control process. Therefore, the implementation of different control objectives in the controller was done through setting the $Q$ matrix appropriately. As such, a Position Control strategy was implemented by setting the weight of cursor position in the $Q$ matrix to a large value, emphasizing the primacy of cursor position as a control variable. Similarly, to implement the Velocity Control strategy, the weight of the cursor velocity in the $Q$ matrix was set to a large value (see Methods). By simulating the task for each control strategy, we could generate synthetic behavior similar to that of humans and monkeys. **Figure 3B and C** illustrate a few example simulations of the task under different difficulty levels for the Position Control and Velocity Control, respectively. As exemplified, the simulated trials for Position Control show oscillatory movements of the cursor around the center, whereas the trials generated based on Velocity Control, exhibited slow drift of the cursor from the center with minimal attempt to correct for such drift. These characteristics were similar to the observed patterns of behavior in human and monkey data (**Figure 1C and D**).

To further identify the behavioral signatures associated with each control objective, beyond the apparent differences between single trials, we conducted a series of simulations in which the model

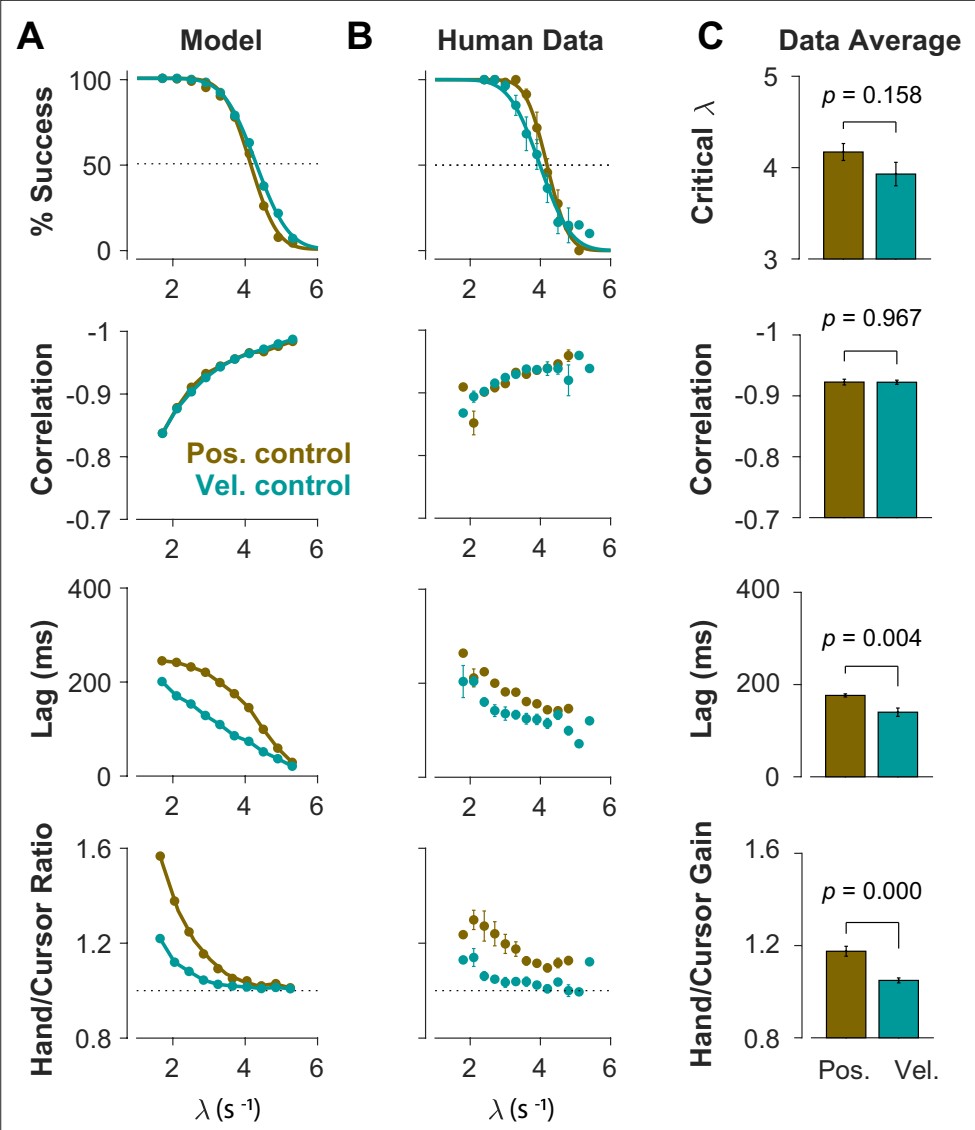

**Figure 4.** Different control objectives result in measurably different behavior. Overall performance of the model (**A**) and human subjects (**B**) for two control objectives, Position Control and Velocity Control. The four rows show success rate (first row), correlation between hand and cursor position (second row), sensorimotor lag between cursor and hand position (third row), and the hand/cursor RMS ratio, defined as the RMS of hand movement over the RMS of cursor movement during each trial (last row). The error bars on the human average data indicate the standard error of the mean across subjects for each group (n=6 per group). (**C**) The average performance across difficulty levels and subjects within each group. The critical $\lambda$ (first row) indicates the difficulty level at which the success rate crosses 50%. The *p*-values are produced using unpaired t-test.

performance was examined for a range of task difficulties. Novel predictions of the model for each control objective were assessed. For each objective, the task was simulated for different difficulty levels, ranging from $\lambda = 1.5$ to $\lambda = 7$, with increments of $\Delta\lambda = 0.2$. For each difficulty level, 500 trials were simulated (see Methods for details). In the first step, we performed the same set of analyses as reported in *Figure 2* to evaluate how the model compared to human and monkey behavior at an aggregate level of CST performance. *Figure 4A* illustrates the overall performance of the model for both Position Control and Velocity Control. As shown, for each metric, the model exhibited comparable behavior to the experimental data with regard to the task difficulty: the success rate dropped in a sigmoidal fashion, the strength of the correlation between hand and cursor position increased, and the response lag between hand and cursor as well as the hand/cursor RMS ratio decreased.

These results showed that, overall, both simulated control objectives produced similar behavioral characteristics as humans and monkeys. More interestingly, the model predicted that Position and Velocity Control performed comparably in success rate and hand-cursor correlation (*Figure 4A*, top two panels), but differed significantly in the response lag and the hand/cursor RMS ratio (*Figure 4A*, bottom two panels). Specifically, Position Control consistently showed larger values for lag and RMS ratio for most task difficulty levels.

### Experiment 2: CST performance under explicit instructions

The model indicated that differences in behavioral metrics existed for Position vs Velocity Control. This led to a new experiment for which we recruited two new groups of human subjects (n=6 per group). Each group performed the CST under the same procedure as described in Experiment 1, except that this time each group was explicitly instructed to use a specific control objective. One group was asked to perform the task with 'keeping the cursor at the center of the screen at all times'. This instruction intended to induce Position Control. The second group was asked to 'keep the cursor still anywhere within the boundaries of the screen'. This instruction aimed to induce Velocity Control (see Methods for details). In each group, the kinematic behavior of hand and cursor was collected, and the control metrics were calculated. The goal was to elicit differences in performance between the two groups and, if such differences were found, to determine whether they matched the behavior of the corresponding model.

The summary of performance for both human subject groups is shown in *Figure 4B*. The general trends of all four measures with respect to the task difficulty were consistent with the data generated by the model, as well as the human data from Experiment 1 (*Figure 2*). Importantly, the behavioral differences between the two control strategies in human data matched the predictions of the model relatively well (*Figure 4A and B*): the rate of success was similar, and with the exception of hand-cursor correlation, the group with Position Control instruction showed a significantly larger hand-cursor lag (unpaired t-test: $t_{10}=3.79$, p=0.004) and hand/cursor RMS ratio (unpaired t-test: $t_{10}=5.27$, $p<10^{-3}$) compared to the group with Velocity Control instructions (*Figure 4C*).

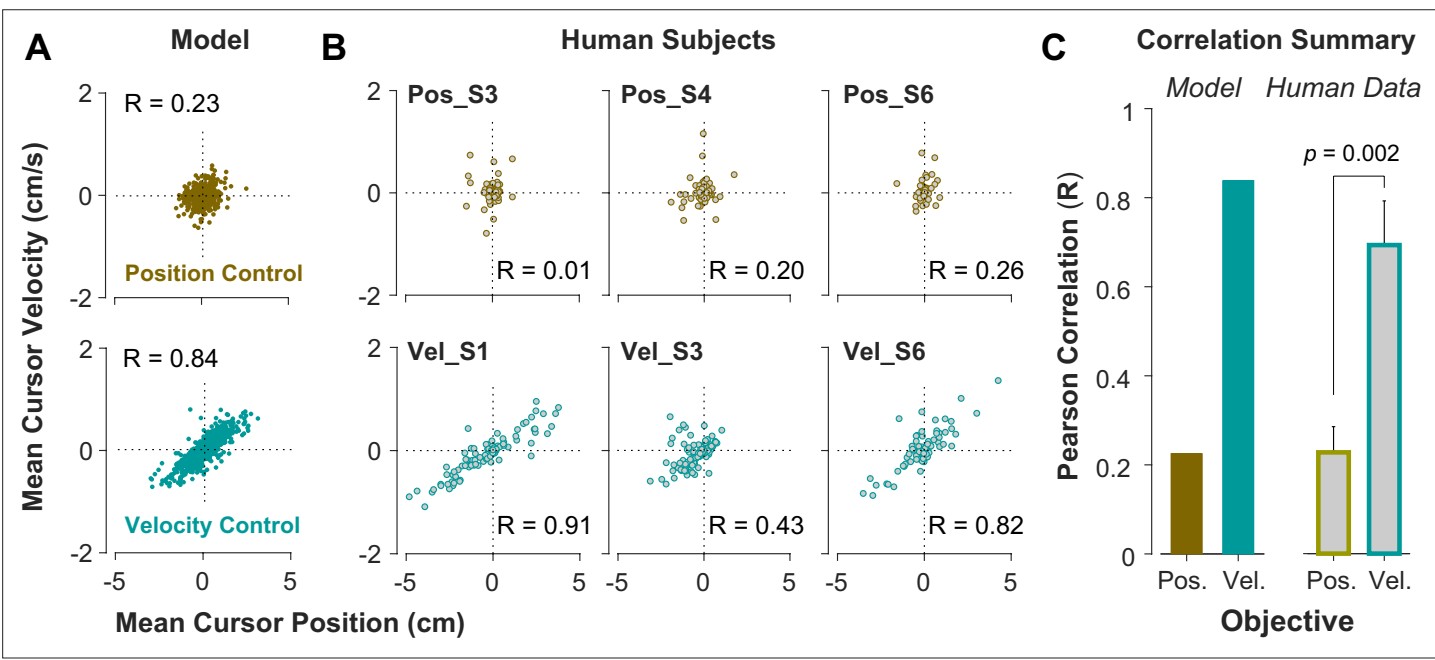

**Figure 5.** State-space distribution of trials reveals different control strategies. (**A**) Mean cursor velocity plotted against mean position for each trial, shown for the position control objective (top) and velocity control objective (bottom). Each data point represents one successful trial and was simulated for a range of difficulty levels up to the critical $\lambda$ value (corresponding to 50% success rate). (**B**) Three example human subjects from the position control group (top row) and velocity control group (bottom row). Each data point represents one successful trial. The data represents an ensemble of trials ranging in difficulty levels up to the critical $\lambda$ value for each subject. R indicates the correlation between the trial position and velocities. (**C**) Pearson correlation coefficient between cursor mean position and velocity for each control objective in the model (left) and human data (right). The human data shows the mean (± SE) across subjects for each control objective group (n=6 per group). The *p*-value is produced using an unpaired t-test.

These results showed that the model not only captured the overall performance features observed in the data, it also successfully demonstrated the redundancy of control strategies in CST performance. Importantly, it could qualitatively distinguish between such strategies at an aggregate level of performance. To ask further, can we identify in a quantitative way the control objective employed by an individual? Can we do so even in a given trial, when no explicit information about their preferred objective was available? To this end, we examined performance at a single-trial level and introduced quantitative measures that evaluated the degree to which a particular control objective was used in that trial, as described in the next section.

## Behavioral traces of control objectives in an individual's overall performance

To further investigate what control objective was preferred by an individual or in a given trial, we examined the predictions of the model about the cursor behavior in state space, and then tested these predictions using experimental data from Experiment 2. Two metrics were defined that captured the state-space behavior of the cursor in each trial. First, we examined the average cursor position and cursor velocity in each trial, represented in the state space of cursor movement. This provided a single data point for each trial in state space, indicating whether on average there was a drift in cursor position and its velocity away from zero ($x = \dot{x} = 0$). It was expected that for Position Control, all trials scattered around the origin of the state space, whereas for Velocity Control, they could deviate from the origin. We also examined whether there was a correlation between the cursor mean position and its mean velocity. This, in essence, was equivalent to the autocorrelation of cursor position, i.e., correlation between mean position and final position. *Figure 5A* illustrates the state-space representation of cursor movement based on model simulations for both Position Control (top) and Velocity Control (bottom), where each data point represents one simulated trial. As shown, the distribution of trials in this space differed markedly between the two control objectives (also see *Appendix 1—figures 2 and 3*). Position Control resulted in a distribution with little correlation between cursor position and its velocity, and closely scattered around the center. In contrast, Velocity Control revealed an elongated distribution with a relatively strong correlation between the cursor position and its velocity.

We further examined whether such distinction in behavior was solely due to a change in the control objective, or whether varying other parameters in the model simulations, such as motor noise, delay, or effort cost, could also generate similar distinctive patterns. These sensitivity analyses showed that different magnitudes of noise and delay only affected the success rates, but no other features, as to be expected. Effort cost also could not account for the observed differences in the above mentioned movement distributions (*Appendix 1—figures 4–6*). Therefore, this robustness to variations in the model parameters allowed us to probe performance based on the underlying control objective.

To validate the model predictions, the same analyses of cursor position and velocity were performed on the empirical data from Experiment 2. *Figure 5B* illustrates three example subjects from the Position Control and Velocity Control groups, and *Figure 5C* shows a summary of how the correlation values differed across control objectives for the model and the empirical data. As shown, overall, subjects in the Velocity Control group showed significantly larger correlations than individuals in the Position Control group (unpaired t-test on the Pearson correlation coefficient: $t_{10}$=4.06, p=0.002). Based on the within-group variability, this allowed us to determine how pronounced a subject executed their respective strategy compared to other subjects in the same group.

It should be noted that *Figure 5B* shows the data for an ensemble of trials ranging in difficulty levels from easy up to the critical $\lambda_c$ values, corresponding to success rates ranging from 100% to 50%. Additional analyses probed whether these behavioral features changed as a function of the difficulty levels. When grouping the trials into easy and moderate difficulty trials, the cursor position and velocity relations did not change and the correlations for the two control objectives continued to reveal the same relative difference between the control objectives (see *Appendix 1—figures 7 and 8*).

## The effects of control objective at a single-trial level of behavior

Due to the task's redundancy, the control objective may not be fixed for an individual throughout their performance and might vary from one trial to the next. It is therefore of great interest to determine, in a given trial, to what extent the behavior is the outcome of Position or Velocity Control. To this end,

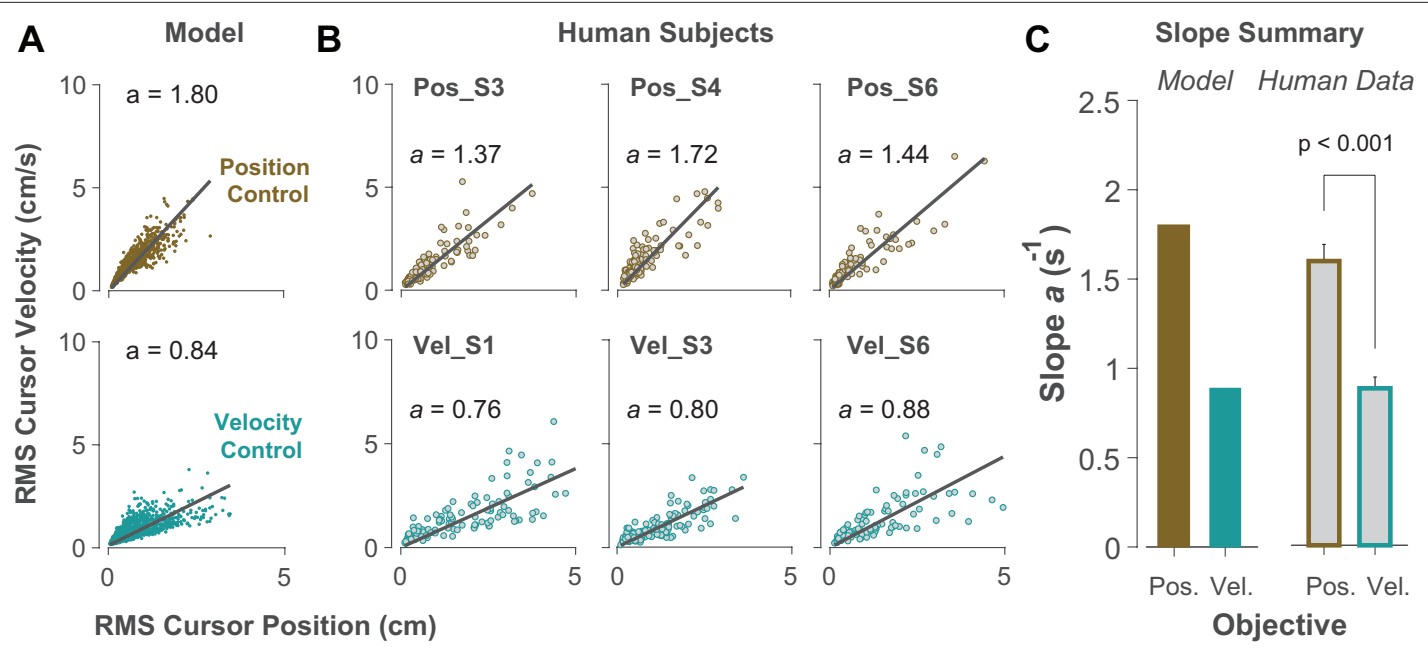

**Figure 6.** Identifying control objective based on magnitude of cursor movement in the state space. (**A**) Magnitude of cursor movements quantified by the RMS of position and cursor velocity for each trial, plotted against each other; position control objective (top) and velocity control objective (bottom). Each data point represents one successful trial and was generated based on the model simulations for a range of difficulty levels up to the critical $\lambda$ value (corresponding to 50% success rate). (**B**) Performance of three example subjects from the position control group (top row) and velocity control group (bottom row). Each data point represents one successful trial. The data represents an ensemble of trials ranging in difficulty level up to the critical $\lambda$ value for each subject. The values of the regression slopes are also shown. (**C**) Summary of the regression slopes for the RMS plots, shown for each control objective in the model (left) and human data (right). The human data shows the mean (± SE) across subjects for each control group (n=6 per group). The *p*-value is produced using an unpaired t-test.

we examined the magnitude of cursor movement calculated as the root mean squared (RMS) of its position and velocity in each trial. This was directly related to the cost functions used in the model (*Equation 2*), which provided a more direct comparison regarding the primacy of Position versus Velocity Control of the cursor: Position Control aimed to minimize the RMS of cursor position, while Velocity Control aimed to minimize the RMS of cursor velocity. This distinction could be well represented in the state-space of the cursor movement.

*Figure 6A* illustrates the model prediction for the RMS of cursor position and cursor velocity plotted against each other for Position Control (top) and Velocity Control (bottom). For Position Control, the distribution of trials leans towards the vertical axis (restricting cursor position but allowing large cursor velocities), whereas for Velocity Control, it leans mainly towards the horizontal axis (a larger range of cursor positions but restricted velocities). This distinction could be quantified by the slope of a fitted regression line to the data, with relatively larger slopes indicating Position Control and smaller slopes signaling Velocity Control. Similar patterns of behavior could be observed in the human data from Experiment 2 as illustrated in *Figure 6B and C*, with the Position Control group showing significantly larger regression slope than the Velocity Control group (unpaired t-test, $t_{10}=6.33$, p<0.001). The regression slope could more clearly distinguish between individual trials than could the correlation coefficient metric shown in *Figure 5*, regarding their corresponding control objective: if a given trial in the RMS space of the cursor movement lay below/above a certain slope threshold, its performance could be considered the result of Velocity/Position Control. We could therefore use this behavioral feature to develop a classifier that inferred, with a certain level of confidence, the underlying control objective in the performance of an individual in any given trial.

## Inferring control objectives from behavior during CST performance

When monkeys performed the CST, we lacked explicit knowledge about which strategy they might have employed. This resembles Experiment 1 when humans performed the CST with no specific

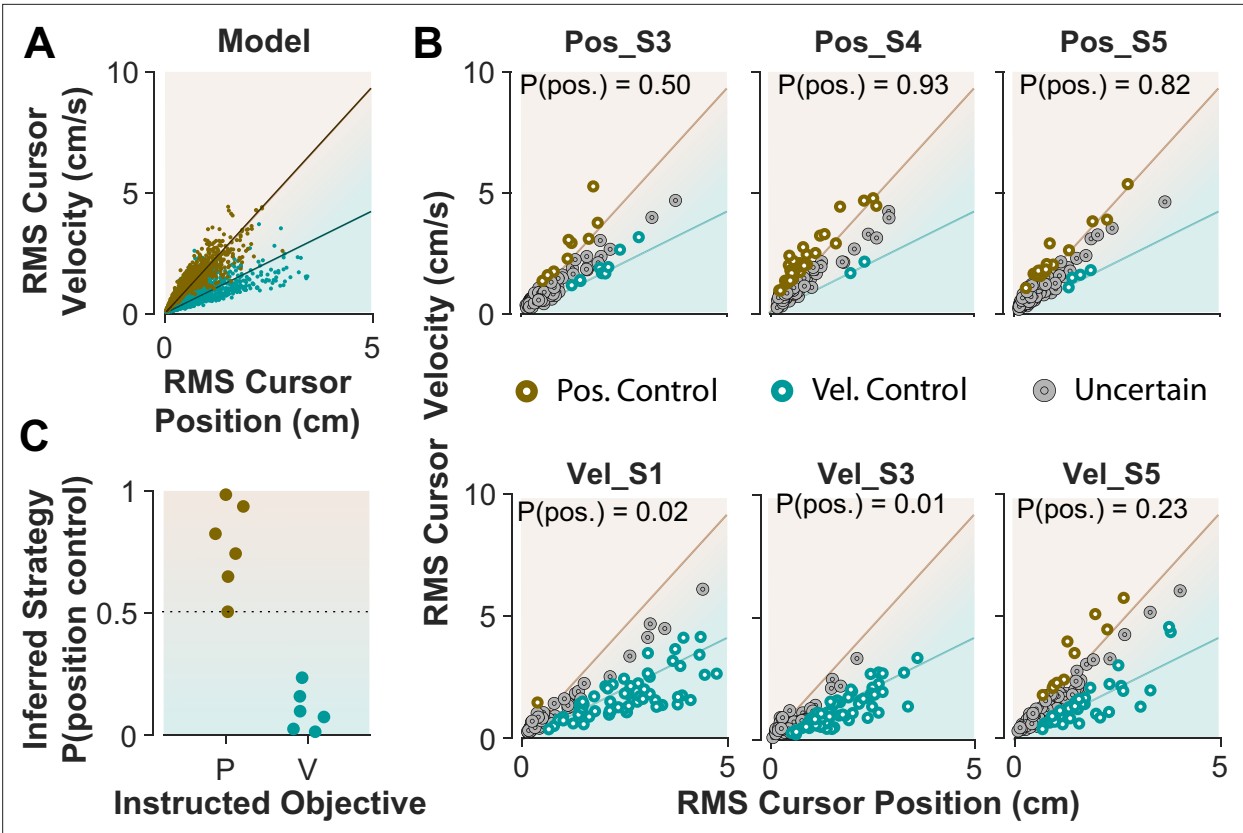

**Figure 7.** Classifying control objectives in humans who received explicit instructions. (**A**) Simulated data in the RMS space of cursor movement used as training set for a classifier to determine the control objective of each trial. (**B**) Data from three example subjects in each group, where each trial was classified as Position Control (brown), Velocity Control (cyan), or Uncertain as to the control objective (grey). To obtain the control objective of each trial, the classifier (a support vector machine; see Methods) obtained the probability of that trial performed with Position Control, where P(pos)>95% was classified as Position Control, P(pos)<5% was classified as Velocity Control, and everything else was classified as uncertain. The average of P(pos) across all trials for each individual is shown inside the respective plot. (**C**) Overall probability of Position Control summarized for all subjects instructed in the position and Velocity Control groups of Experiment 2.

instructions and their control objective was not explicitly available. To achieve the goal of inferring an individual's control objective based on their performance, we used the control characteristics that our computational approach introduced to distinguish between different control objectives. To this end, the simulation results based on the cursor movement in its RMS space (*Figure 6A*) were used to train a simple classifier, a support vector machine (see Methods). This classifier then determined, based on the learned regression slopes from the RMS distributions (*Figure 7A*), whether a given trial was likely performed as Position Control, or Velocity Control.

We first evaluated the accuracy of the classifier based on simulated data, where the classifier was trained on synthetic data generated with Position and Velocity Control (2250 trials each), and then tested on a separate set of trials from either control objective (2500 trials). For any given trial, the classifier obtained a posterior probability indicating to what extent that trial was generated under Position Control. The probability of 95% and higher identified the given trial as Position Control, while the probability less than 5% labeled the trial as Velocity Control; anything in between was considered 'Uncertain' as to the underlying control objective. We also measured how often the classifier misclassified a Position Control trial as Velocity Control, and vice versa. The results showed that 6.2% of trials that were generated with Position Control were misclassified as Velocity Control, and 5.5% of trials that were generated with Velocity Control were misclassified as Position Control. This provided a reasonable accuracy for the classifier applied to the experimental data.

Next, we tested the performance of the classifier on the empirical data from Experiment 2, where the intended control objective used by each subject was known. We asked how well the classifier could recover the control objective used by each subject. *Figure 7B* shows the cursor RMS data from

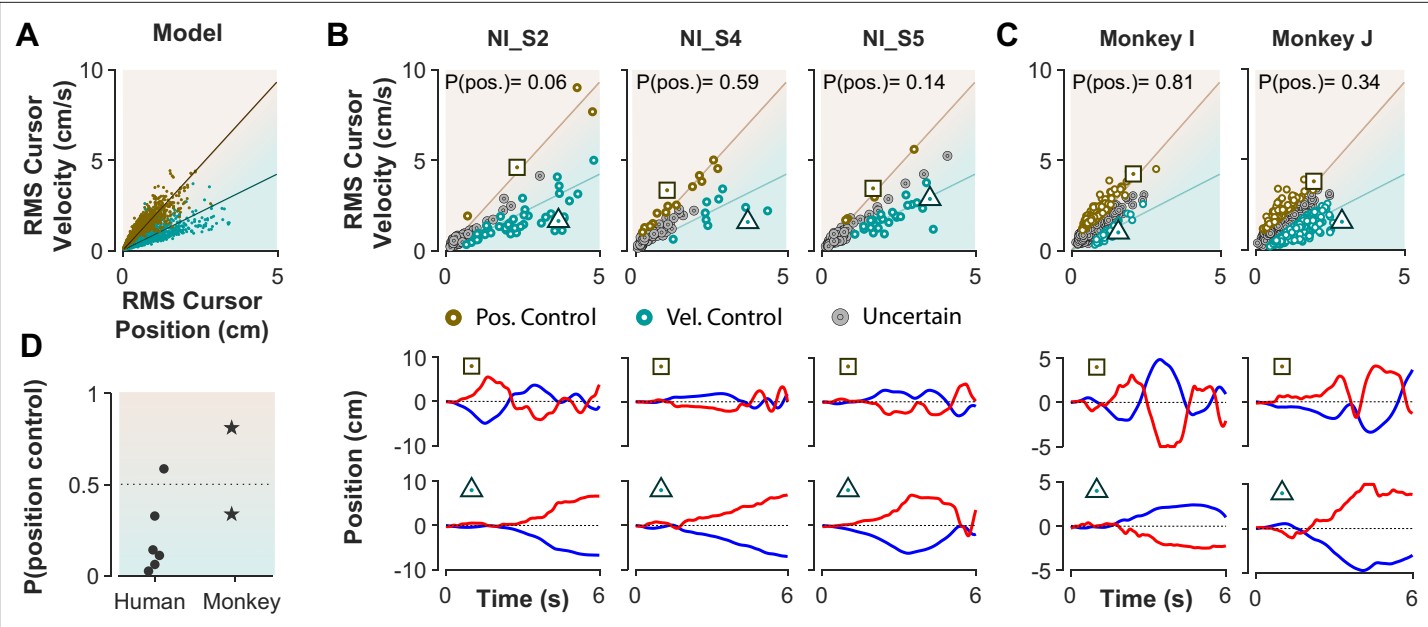

**Figure 8.** Inferring control strategies in monkeys and humans who received no instructions. (**A**) Simulated data in the RMS space of cursor movement was used as training set for a classifier to determine the control objective of a trial without explicit instructions. (**B**) Data from three example human subjects with no instructions (NI) about the control objective. Each trial (data point) is classified based on the probability of Position Control, P(pos), obtained for each trial from the classifier. Trials with P(pos) >95% and P(pos) <5% were, respectively, labeled as Position Control (brown) and Velocity Control (cyan), while other probabilities were labeled as Uncertain (grey). Two example trials, one from each control objective, are shown in the bottom row. (**C**) The classifier was used on data from two monkeys (Monkey I and J) who performed the CST. Similarly, trials for each monkey were categorized as Position Control (brown), Velocity Control (cyan), or Uncertain (grey). (**D**) Overall probability of an individual preferring Position Control, shown for six humans and two monkeys. This measure was obtained for each individual as the average probability of Position Control across all trials.

three example subjects in each instructed group (similar to *Figure 6*). For each trial (data point) the probability of Position Control was estimated by the classifier, and the trial was identified as Position or Velocity Control based on the estimated probability (>95% for Position, and <5% for Velocity Control). As shown in *Figure 7B*, for the Position Control group, most of the trials were rightfully classified as Position Control, and similarly for the Velocity Control group, the majority of trials were classified as Velocity Control. The average probability across all trials for each individual was also obtained as an overall measure of the control objective for that subject. This average measure is shown in *Figure 7B* for the example subjects and summarized in *Figure 7C* for all subjects in each group. This showed that the classifier correctly determined the control strategy of each individual without being trained on any experimental data.

The ultimate test of our approach would be to infer the control strategy used by individuals whose control objective was unknown, that is the monkeys and the humans who received no instructions about the control strategy in Experiment 1. After representing the performance of each subject in the RMS space, the classifier was used to determine what control objective was used in each trial. *Figure 8* illustrates the classification results for human subjects of Experiment 1 as well as two monkeys (*Monkey I and J* from *Quick et al., 2018*). The model simulations are also provided as reference in *Figure 8A*. *Figure 8B and C* show the data from three example human subjects, as well as two monkeys, in which each trial is either labeled as Position Control (brown), Velocity Control (cyan), or Uncertain (grey). Two example trials, one from each inferred control strategy are also singled out from each subject's performance in *Figure 8B and C* (bottom row) to show how the hand and cursor movement behaved under each control objective. Calculating the average probability of control objective for each individual, similar to *Figure 7*, we could infer which control objective was of primary importance for each subject (*Figure 8D*). For example, human subject NI-S2 more likely adopted Velocity Control, while human subject NI-S4 mainly performed the task with Position Control (*Figure 8B*). Similarly, *Monkey I* seemed to prefer Velocity Control, while *Monkey J* most likely adopted Position Control (*Figure 8C*).

Ultimately, our procedure enabled us to not only infer the underlying control objective at a single trial level, but also identify which objective was overall preferred by humans and monkeys when no explicit knowledge about their strategy was available. These results are encouraging as they constitute an important step towards bridging our findings between human and monkey research, and ultimately guide neurophysiological analyses to identify the neural underpinnings of control objectives in the primates' brain.

## Discussion

As we seek to understand the neural basis of human motor control, it is important to build links between studies in humans, where behavior can be complex and naturalistic, and monkeys, where direct neural recordings are possible (*Badre et al., 2015*). Doing so requires close coordination between researchers who work with humans and animals, because this work is typically done in separate labs with a few notable exceptions (*Kurtzer et al., 2008*; *Pruszynski et al., 2011*). With the goal to advance insights into movement control, the current work explicitly paralleled human-monkey behavior in a novel paradigm for monkey research. In a matching task design, humans and monkeys performed a virtual balancing task, where they controlled an unstable system using lateral movements of their right hand to keep a cursor on the screen. The task was challenging and, importantly, exhibited different ways to achieve task success. The task required skill that was nevertheless simple enough for monkeys to learn and ultimately achieve the same level of proficiency as humans.

The results showed that both humans and monkeys exhibited the same behavioral characteristics as the task was made progressively more difficult: success rates dropped in a sigmoidal fashion, the correlation magnitude between hand and cursor position increased, and the response lag from cursor movement to hand response decreased. Further observations based on single trials showed that the task was possibly achieved with different control objectives, both across subjects and across trials. Our goal was to identify the underlying control objectives that led to different behavior; a model based on optimal feedback control was developed that identified two different control objectives that successfully captured the average performance features of humans and monkeys: Position Control and Velocity Control. Both strategies produced behavior that was consistent with observations even at the single trial level. Additional experiments revealed that humans who followed specific instructions as to performing the task with Position Control ('keep the cursor at the center') or Velocity Control ('keep the cursor still') matched the behavior predicted by the two simulated control objectives. Model simulations exhibited features that served to identify these two control objectives in humans and monkeys who received no specific instructions at a single trial level.

Studies in motor neurophysiology have largely relied on simple paradigms such as center-out movements (*Batista et al., 1999*; *Cisek et al., 2003*; *Georgopoulos et al., 1986*; *Pruszynski et al., 2011*; *Scott and Kalaska, 1997*), which were brief in duration, highly stereotypical across repetitions, and could be performed to a reasonable degree of success with limited sensory feedback. Such characteristics were needed to make sense of noisy neural data through averaging trials over many repeats of highly similar behaviors. However, such constrained behaviors are not common in natural settings, where we continually utilize sensory feedback to respond to our environment, interact with objects around us, and never do the same action the exact same way. Indeed, such fluid, prolonged and feedback-driven interactions are what we seek to understand both at the behavioral and neural levels. To this end, we need to investigate more complex tasks that involve sensory-driven control and allow for different control strategies while still within a sufficiently controlled scope. The task employed here, the CST, continuously engages feedback-driven control mechanisms for a prolonged period of time and is rich in its trial-to-trial and subject-to-subject variability. As we can titrate the difficulty of the task, both monkeys and humans can learn it and we can study and model their behavior. This opens the gate towards understanding the neural principles of skill learning beyond simple reaching tasks. This study showed that CST afforded the examination of control strategies through a computational approach that modeled monkey and human behavior in comparable fashion.

Looking across monkey and human behavior is not new per se. In the eye movement literature, comparisons between features in monkeys and humans have been more common (e.g. *Dorris et al., 2000*; *Groh and Sparks, 1996*). In fact, experimental findings in primates have been instrumental to understanding impairments in eye movement control in humans. And yet, due to practical considerations, experiments in non-human primates take considerably longer than human behavioral studies;

due to the logistic problems of having research facilities for both monkeys and humans, combined studies in a single lab have largely remained elusive. Direct comparative studies conducted in different laboratories with exact matches between experimental conditions are harder to achieve.

Another possible avenue for bridging insights between human and monkey behavior is through a computational approach applied to both human and monkey performance (*Badre et al., 2015*; *Rajalingham et al., 2022*). In an earlier attempt of modeling CST, a simple PD controller with delay in sensory feedback was proposed to explain the recorded behavior (*Quick et al., 2018*). However, the model was limited in its ability to capture most features observed in the data, such as success rate or correlation between hand and cursor position. In the past years, optimal feedback control (OFC) has been introduced as an effective approach to understanding the control mechanisms of reaching movements at the level of behavior (*Diedrichsen et al., 2010*; *McNamee and Wolpert, 2019*; *Pruszynski and Scott, 2012*; *Scott, 2004*; *Todorov, 2004*), separately in human research (*Liu and Todorov, 2007*; *Nagengast et al., 2010*; *Nashed et al., 2014*; *Razavian et al., 2023*; *Ronsse et al., 2010*; *Todorov, 2005*; *Todorov and Jordan, 2002*; *Yeo et al., 2016*) and monkey research (*Benyamini and Zacksenhouse, 2015*; *Cross et al., 2023*; *Kalidindi et al., 2021*; *Kao et al., 2021*; *Takei et al., 2021*). Here, the OFC framework was used to account for and make novel predictions about behavioral features in CST.

Note that there are fundamental differences between reaching and CST movements, which needed to be accounted for in the modeling process. Unlike center-out reaching, the CST did not have a stationary target toward which the hand needed to move; rather, it required the hand/cursor to remain anywhere within a predefined area for a prolonged period of time. Also, the behavior was not tracking a point on the screen, but rather moving in opposite direction of the cursor, a behavior that probably requires more cognitive resources. Despite these advanced task features, OFC as a feedback control framework proved an appropriate approach to examine this demanding interactive and sensory-driven task.

A few aspects in our computational approach are worth discussing. First, we examined control objectives that only involved two main kinematic quantities of movement: cursor position and cursor velocity. One might argue that other kinematic features could be explored, such as acceleration or other higher derivatives of the cursor and/or the hand. However, it is important to note that, given the task of keeping the cursor within a specified area for a period of time, cursor position and velocity are the most directly related quantities to the goal of the task. These quantities were also less demanding to predict from sensory feedback, compared to, for example, acceleration (*Hwang et al., 2006*; *Sing et al., 2009*). Also note that the kinematics of the hand were not the variables of interest in the task, as the goal was to control the cursor, and not the hand.

Second, there may indeed exist simpler control models that can exhibit similar distinctions in behaviors simply by finding the right control gains for each behavioral pattern. In essence, our modeling approach also generates such patterns using control gains at the extremes of the spectrum of Position and Velocity Control. However, in contrast to models with a simple gain modulation, our model provides a normative account of what control gains are needed to account for the observed data, and what objectives underlie the choice of such gains (see *Appendix 1—figure 9*, for the optimal gain modulation across control objectives and difficulty levels). In this case, the two control objectives not only demonstrated the ability to generate the distinct behavioral traces of our data, but also accounted for the main performance features such as success rate, lag, RMS ratio, and correlations across a wide range of difficulty levels. And yet, we do not claim that our approach provides a 'ground truth'. Rather, it presents a reasonable account of behavior with an intuitive explanation about human and monkey performance in our virtual balancing task.

Third, we mainly explored Position and Velocity Control separately to identify distinctive behavioral features associated with each one. Experimental data, however, shows that a large number of trials fall somewhere between the Position and Velocity Control boundaries (*Figures 7 and 8*). This could be due to a mixed control strategy, where both Position and Velocity Control contribute simultaneously to achieving the task goal, or where subjects switch strategies of their own accord. Here, we aimed to determine the behavioral signatures of the extreme cases, either predominantly based on position, or velocity of the cursor movement. This may increase the chance to detect differences more clearly in neural activity associated with each control objective in further analysis of monkeys' neurophysiological data.

Even though in this experiment only a subset of trials was amenable to a clear identification as one control strategy, it is possible with monkeys to collect tens of thousands of trials over many days accumulating enough trials for analysis. Furthermore, employing more sophisticated experimental manipulations in future studies, such as introducing perturbations during task performance, could potentially enhance the distinction between control objectives and elucidate their underlying neural mechanisms. This was briefly tested in a series of additional simulations in our study, whereby introducing a simple random offset in the initial cursor position more clearly distinguished between different control objectives (*Appendix 1—figure 10*). Experimental evaluation of these model predictions is left for future studies.

Having identified control objectives in behavior, the next intriguing question is what neural activity could underlie these different behavioral signatures. As our task is quite novel to the field, it is difficult to formulate exact predictions. However, one first step amenable to analysis is how neural activity differs when preparing for the trial. Previous work has shown that the motor cortex is highly active prior to an action and neural dynamics become specific to the task as monkeys prepare for a cued movement (*Ames et al., 2014*; *Cisek and Kalaska, 2005*; *Dekleva et al., 2018*; *Elsayed et al., 2016*; *Kaufman et al., 2014*; *Lara et al., 2018*; *Perich et al., 2018*; *Vyas et al., 2018*; *Zimnik and Churchland, 2021*). It seems possible that the control objectives we observed elicit different preparatory activity in the motor cortex.

To conclude, despite potential limitations, our approach was successful in two main ways. First, it provided a normative explanation for the macro-level characteristics of behavior observed in human and monkey data. Second, due to its generative nature, model simulations also provide for not yet seen conditions and can make predictions about the behavior under new control objectives. Hence, our behavioral analysis holds promise to generate crucial insights into neural principles of skillful manipulation, not only in monkeys but also, by induction, in humans.

## Methods

### Participants and ethics statement

18 healthy, right-handed university students (age: 18 to –25 years; 8 females) with no self-reported neuromuscular pathology volunteered to take part in the experiments. All participants were naïve to the purpose of the experiment and provided informed written consent prior to participation. The experimental paradigm and procedure were approved by the Northeastern University Institutional Review Board (IRB# 22-02-15).

The data from two adult male Rhesus monkeys (*Macaca mulatta*, wild type, supplied by Alpha Genesis, Ages: 7 and 8 years old) used in this study was taken from a previously published work (*Quick et al., 2018*). All animal procedures were approved by the University of Pittsburgh Institutional Animal Care and Use Committee, in accordance with the guidelines of the US Department of Agriculture, the International Association for the Assessment and Accreditation of Laboratory Animal Care, and the National Institutes of Health. For details of experimental rig and procedure see the Methods in *Quick et al., 2018*.

### Critical stability task (CST)

The CST involved balancing an unstable cursor displayed on the screen using the movement of the hand (*Jex et al., 1966*; *Quick et al., 2014*; *Quick et al., 2018*). The CST dynamics was governed by a first-order differential equation as shown in *Equation 1*. The difficulty of the task was manipulated by changing the parameter $\lambda$: by increasing $\lambda$ the task became more unstable, hence more difficult to accomplish. To perform the task, subjects sat on a sturdy chair behind a small table, with their right hand free to move above the table (*Figure 1*). A reflective marker was attached to the subject's back of the hand on the third metacarpal bone. The hand position was recorded using a 12-camera motion capture system at a sampling rate of 250 Hz (Qualisys, 5+, Goetheburg, SE). The mediolateral component of the hand position was used to solve the CST dynamics with the initial condition of $x(t=0)=0$ (*Quick et al., 2018*). The calculated cursor position was real-time projected as a small blue disk (diameter: 4 mm, approximately 0.8° in visual angle) on a large vertical screen in front of the subject at a 150 cm distance. The processing delay of the visual rendering was roughly 50 ms.

## Experimental design

### Task

At the beginning of the experiment, human subjects held their right hand comfortably above the table and in front of their right shoulder as shown in *Figure 1*, where the hand position was mapped to the center of the screen. The visual display of the cursor and hand position was scaled such that the lateral hand movements of ±10 cm corresponded to ±20° of visual angle from the screen center and served as the boundaries of the workspace. Each trial started with the hand position displayed on the screen as a red cursor (diameter: 4 mm, or approximately 0.8° in visual angle). Subjects were asked to bring the red cursor to the center of the screen depicted by a small grey box (*Figure 1*). Once the red cursor was at the center, and after a delay of 500 ms, the trial started. The red cursor disappeared and a blue cursor representing the $x$ position in *Equation 1* appeared at the center. Subjects were instructed to keep (or 'balance') the blue cursor within the boundaries of the workspace for 6 s for the trial to be considered successful. If the cursor escaped the workspace at any time, the trial would abort and considered as failed. Subjects were informed of the outcome of the trial by a message on the screen, reading 'Well Done!' for success, and 'Failed!' for failure. This feedback matched the binary reward that monkeys were given in the experiment by Quick and colleagues. The next trial started after an intertrial interval of 1000 ms.

## Experimental paradigm and conditions

Each human subject participated in the experiment for three consecutive days. At the beginning of the first day, subjects were familiarized with the experimental setup and the objectives of the task. Familiarization consisted of five CST trials with moderate difficulty level. These trials were later excluded from the analyses. The main experiment consisted of three main phases that were repeated on each day. The first and second phases of the experiment involved 15 reaction time trials and 10 tracking trials, respectively (data for reaction time and tracking trials were not reported in this study). Phase three involved the CST trials, which were performed in three blocks. In Block 1, subjects performed 30 CST trials, where the difficulty level was determined in each trial using an up-down method: starting from $\lambda = 2.5$ in the first trial, if subjects succeeded/failed on the current trial, $\lambda$ was increased/decreased by $\Delta\lambda = 0.2$ in the next trial. By the end of Block 1, subjects had gradually converged to $\lambda$ values in which the success rate was approximately 50%. This value was considered as the critical instability value (*Quick et al., 2018*), denoted by $\lambda_c$, and was obtained by averaging the $\lambda$'s of the last five trials of Block 1.

In Block 2, a stepwise increase in $\lambda$ was adopted: subjects started with a difficulty level of $\lambda = 70\%\lambda_c$ (using $\lambda_c$ from the previous block). They continued until they completed 10 successful trials, or 20 trials in total (whichever occurred first). The difficulty level was then increased by $\Delta\lambda = 0.2$, and the procedure repeated. This incremental increase of $\lambda$ continued until the subjects' success rate for the ongoing $\lambda$ dropped below 10% (i.e., less than 2 successful trials out of 20). This marked the end of the second block. In total, subjects performed approximately 120–200 trials in Block 2, depending on the individual's performance.

In Block 3, subjects performed the CST under three selected difficulty levels of easy, medium, and hard, with 20 trials for each difficulty level. These levels corresponded to $\lambda$ values that led to 75% success rate (easy), 50% success rate (medium) and 25% success rate (hard) obtained from each individual's performance in Block 2. The exact values of $\lambda_{75\%}$, $\lambda_{50\%}$, and $\lambda_{25\%}$ were calculated by fitting a psychometric curve to the success rate data from Block 2 as a function of $\lambda$ (see *Figure 2*). The order of difficulty was pseudo-randomly selected for each subject. For this study, we only analyzed the CST data from Block 2 (stepwise increase in $\lambda$) as it matched the procedure used in the monkey experiment (*Quick et al., 2018*). Subjects repeated the same experimental procedure on Day 2 and 3.

Three groups of human subjects participated in the experiment, where each group received different instructions about the task goal. The first group was instructed to perform the CST 'without failing to the best of their ability' (no-instruction group); the second group was instructed to 'keep the cursor at the center of the screen at all times' (Position Control group); and the third group was instructed to 'keep the cursor still anywhere within the bounds of the screen' (velocity control group).

## Analysis

To evaluate the overall performance of humans and monkeys during the CST, four quantities were calculated: success rate, correlation between hand and cursor position, hand-cursor lag, and hand/cursor RMS ratio. For each individual, the quantities were calculated as the average across trials for each bin of $\lambda$ values (bin size: 0.3, starting from $\lambda = 1.5$).

The success rate was obtained as the percentage of successful trials within each $\lambda$ bin. A psychometric curve (a Gaussian cumulative distribution function) was then fitted to the success rate data as a function of $\lambda$ to estimate $\lambda_c$ (critical stability, where success rate was 50%):

$$\% \ Success = 50 \left[ 1 - erf \left( \frac{\lambda - \lambda_c}{\sqrt{2}\sigma} \right) \right] \tag{3}$$

where, 'erf' indicates the error function, and $\sigma$ denotes the standard deviation of the Gaussian cumulative. The correlation and lag quantities (*Figure 2*, B and C) were obtained by first cross-correlating the hand and cursor position trajectories in each trial, and then finding the peak correlation and the corresponding lag (*Figure 2*, see also *Quick et al., 2018*). The hand/cursor RMS ratio (*Figure 2*, D) was defined as the ratio of the root mean squared (RMS) value of hand position over the RMS value of the cursor position in each trial.

Finally, to perform the classification analysis used in *Figure 7* and *Figure 8*, a Support Vector Machine method was applied to learn the two-class control objective labels. In order to build and train a classifier, we used 'fitcsvm.m' function in MATLAB, where synthetic data (RMS of cursor position and cursor velocity) was used as training set. To classify experimental data using the trained classifier, the MATLAB function 'predict.m' was used. Finally, the posterior probabilities over each classification (i.e. the confidence on classification) was calculated using the 'fitPosterior.m' function in MATLAB.

## Optimal feedback control model

An optimal control model was used to build control agents that performed the CST with different control strategies. The model involved an optimal feedback controller that moved the hand, a point mass of $M$=1 kg, through a simple muscle-like actuator (*Todorov, 2005*; *Todorov and Jordan, 2002*). The muscle model was approximated by a first-order low-pass filter that generated forces on the hand in the lateral direction as in *Equations 4; 5*:

$$\tau \dot{F} = -F + u \tag{4}$$

$$\ddot{p} = \frac{1}{M} F \tag{5}$$

where $F$ is the actuator force acting on the hand, $\tau$ is the time constant of the low-pass filter, $u$ is the control input to the muscle, and $\ddot{p}$ is the second derivative of the hand position. Combined with *Equation 1*, the model includes cursor position $x$, hand position $P$, and actuator force $F$ as the states of the system. By taking the first derivative of *Equation 1*, the hand and cursor velocity are also included in the state space of the system:

$$\ddot{x} = \lambda \left( \dot{x} + \dot{p} \right) \tag{6}$$

Finally, by combining *Equations 1; 6*, the CST dynamics can be derived as follows:

$$\ddot{x} = \lambda^2 x + \lambda^2 p + \lambda \dot{p} \tag{7}$$

Note that by taking the higher derivative of CST dynamics in *Equation 6*, we practically made the cursor velocity $\dot{x}$ available to the controller as a state of the system, which allowed us to explore different control strategies directly related to the cursor velocity. This was done with the caveat that the initial conditions of the resultant CST dynamics in *Equation 7* should always satisfy *Equation 1*.

In this case, the behavior of the system could be represented by the state vector, $\mathbf{x} = \left[ x, x, p, p, F \right]$, using the state-space form of the system dynamics as shown below:

$$\dot{\mathbf{x}} = A\mathbf{x} + Bu \tag{8}$$

where $A$ and $B$ represent the dynamics of the system, and $u$ is the control input:

$$A = \begin{bmatrix} 0 & 1 & 0 & 0 & 0 \\ \lambda^2 & 0 & \lambda^2 & \lambda & 0 \\ 0 & 0 & 0 & 1 & 0 \\ 0 & 0 & 0 & 0 & 1/M \\ 0 & 0 & 0 & 0 & -1/\tau \end{bmatrix}$$

$$B = \begin{bmatrix} 0 & 0 & 0 & 0 & 1/\tau \end{bmatrix}^T$$

(9)

In order to implement the feedback controller, the state-space equations were first discretized using the time steps of $\delta = 10$ ms. Further, three noise terms were included in the system dynamics representing the motor additive noise $\xi$, signal dependent noise $\varepsilon$ and sensory additive noise $\omega$, according to the models of biological systems (*Harris and Wolpert, 1998*; *Todorov, 2005*). The resultant equations of the system dynamics were presented as shown below:

$$\mathbf{x}_{t+1} = A_d\mathbf{x}_t + B_d(1 + \varepsilon_t C)u_t + \xi_t$$

$$\mathbf{y}_t = H\mathbf{x}_t + \omega_t$$

(10)

where $\varepsilon_t$, $\xi_t$, and $\omega_t$ are zero-mean Gaussian noise term, $C$ is the signal-dependent noise scalar, $\mathbf{y}_t$ represents the sensory feedback, and matrix $H$ determines the available sensory feedback from the vector of states. For our simulations, all the states were available as feedback, therefore, we considered $H = diag\left([1, 1, 1, 1, 1]\right)$. The matrices $A_d$ and $B_d$ were modified according to *Equation 9* for discrete-time representation of the system: $A_d = I + \delta A$ and $B_d = \delta B$, where $I$ was an identity matrix.

Given the overall dynamics of the system, the feedback controller aimed to calculate the optimal motor command $u_t$ based on the sensory feedback $\mathbf{y}_t$ by minimizing the cost function $J$ (*Todorov, 2005*):

$$u_t = argmin(J)$$

$$J = \sum_{t=1}^{n} \left(\mathbf{x}_t^T Q\mathbf{x}_t + u_t^T U u_t\right)$$

(11)

where $n$ was the number of time samples throughout the movement, and $Q$ and $U$ determined the contribution of state accuracy and effort in the cost function, respectively. In all simulations, $U = 10$. The matrix $Q$, however, was appropriately manipulated to implement different state-dependent control strategies (see below). Accordingly, the optimal control law was obtained in the form:

$$u_t = -L_t\hat{\mathbf{x}}_t$$

(12)

where $L_t$ was the optimal control gain that was solved recursively by minimizing the cost function $J$ (see equation 4.2 of *Todorov, 2005* for detailed calculation of $L_t$). Also, $\hat{\mathbf{x}}_t$ represented the *estimated* states of the system based on the provided feedback $\mathbf{y}_t$, which were obtained using a state estimator as shown below:

$$\hat{\mathbf{x}}_{t+1} = \left(A_d - B_dL_t\right)\hat{\mathbf{x}}_t + K_t\left(\mathbf{y}_t - H\hat{\mathbf{x}}_t\right)$$

(13)

Here, $K_t$ was the filter gain matrix which was calculated in a recursive procedure along with the control gains (see equation 5.2 of *Todorov, 2005*).

## Implementing position control

The aim of the Position Control strategy was to maintain the cursor at the center of the screen throughout the trial. This was implemented by penalizing the deviation of the cursor position $x$ from the center. In this case, the matrix $Q$ was set to $Q = diag\left([q, 0, 0, 0, 0]\right)$, where $q \gg 0$ was a constant. As such, the cost of deviation from the center for the cursor position was dominant represented in the value $J$ of the cost function, making the regulation of cursor position at the center, the primary goal of control.

## Implementing velocity control

The Velocity Control strategy aimed to keep the cursor still at any point within the boundaries of the workspace. In this case, upon deviation of the cursor from the center, the main goal was to bring

the cursor to a stop regardless of the location. This was implemented through penalizing the cursor velocity $\dot{x}$ by setting the matrix $Q = diag\left(\left[0, v, 0, 0, 0, 0\right]\right)$, where $v \gg 0$ was a constant.

## Simulations

Given a control strategy, the model generated 500 trials of CST for each level of task difficulty from $\lambda = 1.5$ to $\lambda = 7$, with increments of $\Delta\lambda = 0.2$. The parameters of the hand and the muscle model (*Equations 4; 5*) were fixed to $M = 1$ kg and $\tau = 0.06$ s. A sensory delay of 50 ms was considered when simulating the task with the optimal feedback controller (*Cluff et al., 2019*; *Todorov, 2005*). To implement the delay, system augmentation was used by adding the states from the current time step with all the states from the 5 preceding time steps (*Crevecoeur et al., 2019*; *Todorov, 2004*). The signal-dependent noise terms were set to $\varepsilon_t \sim N\left(0, 1\right)$ and $C = 1.5$. The motor noise was $\xi_t \sim N\left(0, \Sigma\right)$, where $\Sigma = 0.4 BB^T$. For each trial, the simulation started from the initial condition of $x = 0$, and ran for 8 s. Only the first 6 s of each simulation were considered in the analysis for consistency with the experimental paradigm. The success or failure in each simulated trial was decided post-hoc, by determining whether the cursor position $x$ exceeded the limits of the workspace (±10 cm from the center) within the 6 s duration of the trial.

## Acknowledgements

This research was funded by the National Institute of Health R01-CRCNS-NS120579, awarded to Dagmar Sternad and Aaron Batista. Dagmar Sternad was also supported by NIH-R37-HD087089 and NSF-M3X-1825942. Aaron Batista and Patrick Loughlin were also supported by NIH-R01-HD0909125.

## Additional information

### Funding

| Funder | Grant reference number | Author |
|---|---|---|
| National Institutes of Health | R01-CRCNS-NS120579 | Aaron P Batista<br>Dagmar Sternad |
| National Institutes of Health | R37-HD087089 | Dagmar Sternad |
| National Science Foundation | M3X-1825942 | Dagmar Sternad |
| National Institutes of Health | R01-HD0909125 | Aaron P Batista<br>Patrick J Loughlin |

The funders had no role in study design, data collection and interpretation, or the decision to submit the work for publication.

### Author contributions

Mohsen Sadeghi, Conceptualization, Data curation, Formal analysis, Visualization, Writing – original draft, Writing – review and editing; Reza Sharif Razavian, Conceptualization, Data curation, Formal analysis, Writing – original draft, Writing – review and editing; Salah Bazzi, Conceptualization, Formal analysis, Writing – review and editing; Raeed H Chowdhury, Conceptualization, Data curation, Writing – review and editing; Aaron P Batista, Dagmar Sternad, Conceptualization, Supervision, Funding acquisition, Project administration, Writing – review and editing; Patrick J Loughlin, Conceptualization, Supervision, Funding acquisition, Visualization, Project administration, Writing – review and editing

### Author ORCIDs

Mohsen Sadeghi (iD) https://orcid.org/0000-0003-2573-146X
Reza Sharif Razavian (iD) http://orcid.org/0000-0003-1190-0816
Salah Bazzi (iD) http://orcid.org/0000-0002-8631-0426
Aaron P Batista (iD) http://orcid.org/0000-0002-1719-0061
Dagmar Sternad (iD) https://orcid.org/0000-0001-9318-2920

## Ethics

All participants were naïve to the purpose of the experiment and provided informed written consent prior to participation. The experimental paradigm and procedure were approved by the Northeastern University Institutional Review Board (IRB# 22-02-15).

All animal procedures were approved by the University of Pittsburgh Institutional Animal Care and Use Committee, in accordance with the guidelines of the US Department of Agriculture, the International Association for the Assessment and Accreditation of Laboratory Animal Care, and the National Institutes of Health. For details of experimental procedures see the Methods in (Quick et al., 2018).

Reviewer #1 (Public Review): https://doi.org/10.7554/eLife.88514.3.sa1
Reviewer #3 (Public Review): https://doi.org/10.7554/eLife.88514.3.sa2
Author response https://doi.org/10.7554/eLife.88514.3.sa3

## Additional files

### Supplementary files

• MDAR checklist

### Data availability

All data and scripts have been deposited in Dryad: https://doi.org/10.5061/dryad.p2ngf1vzt.

The following dataset was generated:

| Author(s) | Year | Dataset title | Dataset URL | Database and Identifier |
|-----------|------|---------------|-------------|------------------------|
| Salah B | 2024 | Kinematic data of humans performing the critical stability task | https://datadryad.org/stash/dataset/doi:10.5061/dryad.p2ngf1vzt | Dryad Digital Repository, 10.5061/dryad.p2ngf1vzt |

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

# Appendix 1

## Subject-to-subject variability of hand/cursor RMS ratio

In *Figure 2*, a seemingly deviant behavior was observed for the hand/cursor RMS ratio of monkey J compared to the other monkey and the human subjects. This could be due to subject-to-subject variability. Given that data from only two monkeys were available, we examined this possibility by presenting the individual hand/cursor RMS ratio for all individual human subjects. *Appendix 1—figure 1* shows that there was indeed variability across subjects, with some not exhibiting a clear trend with task difficulty. However, on average, the RMS ratio showed a slight decrease as trials grew more difficult, as was earlier shown in *Figure 2*.

## Alternative metrics for inferring control objectives from behavior

We used two main metrics in our analysis of inferring the control objective from behavior based on cursor movement, namely, the mean and RMS of cursor position/velocity (*Figures 5 and 6*). In *Figure 5*, we demonstrated that the choice of control objective affected the correlation between cursor mean position and cursor mean velocity in the state space of cursor movement. As one of the reviewers observed, since the cursor mean velocity over a trial determined the cursor final position in that trial, one could interpret the correlation between cursor mean velocity and its mean position in terms of the autocorrelation function (acf) of cursor position.

In particular, under Position Control, the final cursor position would be relatively uncorrelated with the average position, and hence the temporal acf of position would be narrow. In contrast, under Velocity Control, the final position tends to be similar to the average position, and thus the acf would be wider. We explored this insight by calculating the width of the acf of cursor position for 200 simulated trials at four different $\lambda$ values for each control objective. *Appendix 1—figure 2A* shows example cursor and hand traces, together with the corresponding cursor acf and its width (defined as the width of a rectangle with area equal to the area under the absolute value of the acf). In *Appendix 1—figure 2B*, the distribution of acf width across trials was compared between Position Control and Velocity Control for four example $\lambda$ values. As expected, given the relation between mean and final position under the two control objectives, the distributions of acf widths separate between the two control objectives. As such, acf width could be another metric to dissociate between different control objectives. However, there was a similar overlap between the two objectives, resulting in similarly 'undecided' trials as the metrics we used.

Yet another alternative behavioral metric that could potentially differentiate between different control objectives is one that includes both hand and cursor movement, such as the hand/cursor RMS ratio. *Appendix 1—figure 3* shows the distribution of hand/cursor RMS ratio across simulated trials, generated based on Position Control or Velocity Control for different $\lambda$ values. As shown, this metric also demonstrated the separation of control objectives, albeit with dependence on $\lambda$: as the task difficulty increased, the distributions began to converge, thereby becoming less distinguishable (this effect could also be observed in *Figure 4A*).

Overall, these alternative metrics also reflected the distinction between control objectives in behavior. While they did not offer any observable improvement over our previously used metrics (shown in *Figures 5 and 6*), it is possible that a more exhaustive examination of behavioral features could lead to metrics that better discriminate between control objectives. Such an investigation is beyond the scope of this study.

## Sensitivity analysis of model parameters

We further investigated whether the distinction into two behavioral patterns could also be accounted for by changing other model parameters, specifically the relative cost of effort, motor noise, or sensory delay. To this end, we conducted a series of simulations wherein the control objective remained fixed at either Position or Velocity Control, but effort cost ($U$ in *Equation 2*), noise magnitude ($\epsilon$ in *Equation 10*) and sensory delay were varied independently.

We first examined whether changing the effort cost under a fixed control objective could account for the variability of behavior across groups in Experiment 2. For each control objective, the effort cost was varied between $U$=10, $U$=100, and $U$=1000, and the resulting change in behavior was examined. As shown in *Appendix 1—figure 1A*, the overall performance within a given control objective remained independent of effort cost. In particular, effort cost did not affect the distributions

of cursor mean (*Appendix 1—figure 4B*) and cursor RMS (*Appendix 1—figure 4C*), indicating that the distinctive patterns observed in Experiment 2 could not be explained solely by changing the effort penalty.

Changing the sensory delay time (from 30 ms to 70 ms; *Appendix 1—figure 5*) did impact the success rate at a given $\lambda$, not unexpectedly. However, it did not affect the lag, correlation, RMS ratio, or the distributions of cursor mean and cursor RMS. Changing the level of motor noise (from 10% reduction to 10% increase in noise standard deviation; *Appendix 1—figure 6*) likewise impacted the success rate, but not the other metrics. Overall, these results demonstrated that different control behaviors in the data were predominantly explained by varying the control objective and not effort cost, noise level, or sensory delay.

## Effect of task difficulty on control objectives

We examined whether and to what extent subjects used the same control objective in different task difficulty levels ($\lambda$ values). We only examined this question in subjects who were instructed to adopt a given objective, Position or Velocity Control. *Figures 5–8* presented the data for an ensemble of $\lambda$ values, ranging up to the critical $\lambda$ value ($\lambda_c$ ; associated with 50% success rate). Here, we replot *Figure 5* by separating the trials into two clusters based on task difficulty, Easy, and Moderate $\lambda$ values. The figure shows that the relative behavioral difference between the two control objectives remains qualitatively the same across difficulty levels. Specifically, *Appendix 1—figure 7A* shows the joint distribution of cursor mean velocity against cursor mean position for Easy ($\lambda \leq 70\%\lambda_c$), and Moderate ($70\%\lambda_c \lambda \leq \lambda_c$) conditions. The data are presented for two example subjects, S4 from the Position Control instruction group (brown) and S1 from the Velocity Control group (cyan). As shown, the structure of the data distribution remains approximately the same across Easy and Moderate difficulties, and this was also true with the model simulations (*Appendix 1—figure 7B*). Importantly, the relative structural difference between the two control objectives, quantified by the correlation coefficient between cursor velocity and cursor position $R$, remains unchanged across different difficulty levels as shown in *Appendix 1—figure 7C*. In all cases, $R$ was larger for Velocity Control, indicating consistency in control objective across $\lambda$ values.

The effect of time, or practice, on the control objective as subsumed in the analysis above because subjects performed the task progressing from easy to difficult trials: easy trials were performed early in the experiment, and they became increasingly more difficult towards the end of the experiment. To better examine the time course of possible changes in the control objective, we calculated the probability with which a given trial was performed under the Position Control objective. This probability was obtained from the classifier as applied to each trial. *Appendix 1—figure 8* shows this probability over the course of trials for each individual in each instruction group. As shown, though the trends were noisy, the probabilities remained generally higher for the Position Control group, and lower for the Velocity Control group as expected. Mainly, subjects' performance generally remained within the bounds of their instructed control objective throughout the course of experiment.

## Optimal control gains under different control objectives

The optimal feedback controller in our approach calculates the optimal gains that minimize the cost function (*Equation 2*) for a given control objective. The resulting control command was $u = -Lx$ (*Equation 12*), where $x$ was the state vector and $L$ was the gain vector. Because the cost function depended on the control objective as well as the system dynamics (specifically, the value of $\lambda$), the optimal position and velocity gains would likewise depend on $\lambda$ as well as the control objective. *Appendix 1—figure 9* illustrates the control gains for each cursor state for Position Control and Velocity Control across a range of $\lambda$ values. As shown, the optimal gains under the same control objective varied with task difficulty, that is $\lambda$. This indicates that as $\lambda$ changes, the gains also needed to change in order to (optimally) meet the control objective. Importantly, the choice of control objective was strongly reflected in the cursor position gain, where the two control objectives showed opposing trends across task difficulties.

## Perturbation simulations

To explore the potential for perturbation experiments to enhance the ability to discriminate between control objectives, we implemented a random cursor jump (left or right of screen center) at the start of each trial in 1000 simulation trials of Position Control and of Velocity Control over a range of difficulty levels. The magnitude of the cursor displacement was randomly sampled from a

normal distribution with zero mean (corresponding to screen center) and a standard deviation of 1 cm. *Appendix 1—figure 10A* shows the simulation results for success rate, hand-cursor lag and correlation, and the hand/cursor RMS ratio. As shown, despite the similarity of the success rates for both control objectives, the other metrics showed more pronounced differences between the control objectives, compared to the unperturbed simulations (i.e., *Figure 4*). Interestingly, the difference was more systematic when looking at the cursor states in the mean or RMS spaces. *Appendix 1—figure 10B* shows the joint distribution of cursor mean position and mean velocity, where the different control objectives showed opposite correlations between cursor position and velocity. Similarly, *Appendix 1—figure 10C* showed greater separation between the RMS distributions in Position and Velocity Control.

These simulations demonstrated the potential to more robustly differentiate between different control objectives at the behavioral level, and consequently allowed for clearer parsing of the neural data to search for neural correlates of control objectives. However, the experimental assessment of these predictions is left for future studies.

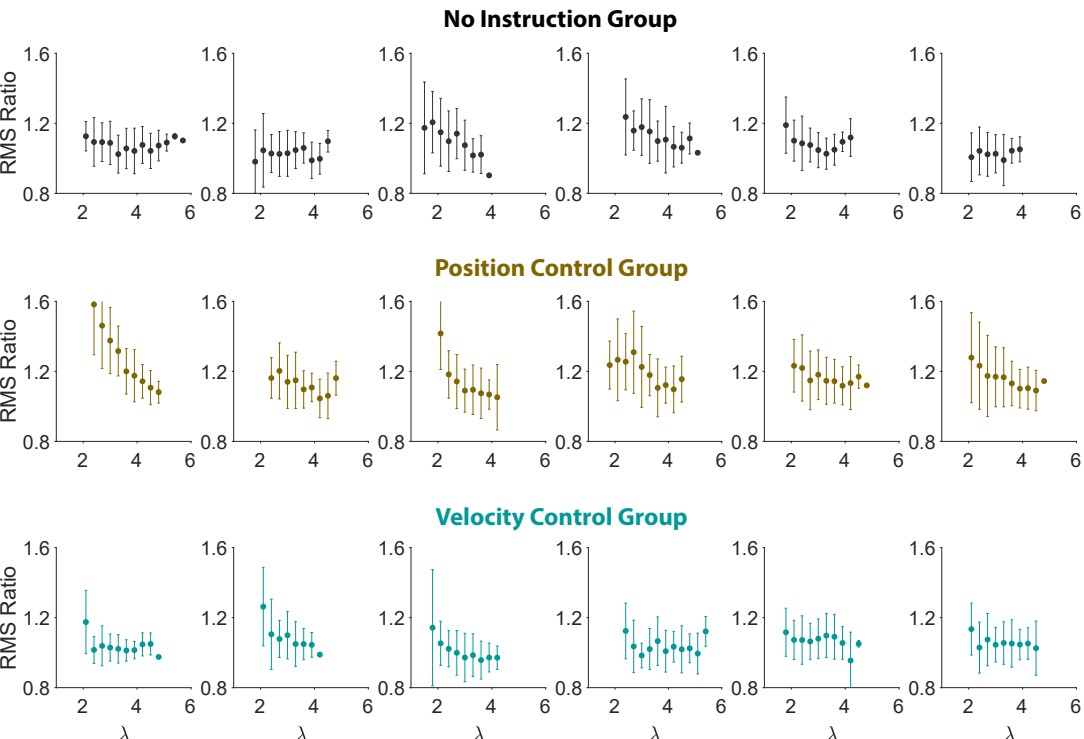

**Appendix 1—figure 1.** Hand/Cursor RMS ratio for individual participants in all three groups (n=6 per group):. **Top**: No Instruction group, **Middle**: Position Control group, and **Bottom**: Velocity Control group. The error bars indicate the standard deviations (SD) across trials for each difficulty level.

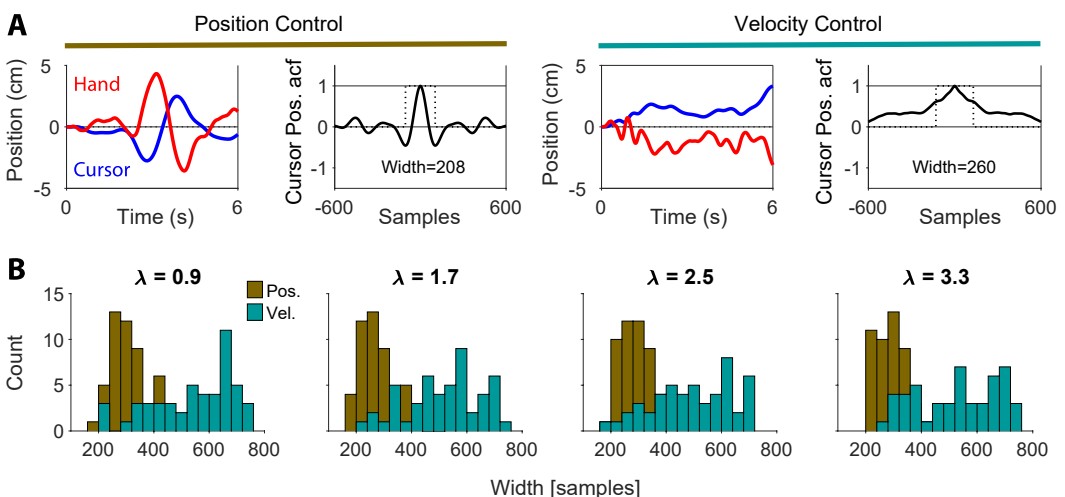

**Appendix 1—figure 2.** Autocorrelation analysis. (**A**) Sample trials with autocorrelation functions (acf) of cursor position shown for position control (columns 1 and 2), and velocity control objectives (columns 3 and 4). The dotted rectangle in the acf plots shows the acf width (see text for definition). (**B**) Histograms of the acf width shown for Position (brown) and Velocity (cyan) control objectives. Each panel shows the results for a different value of $\lambda$ as indicated.

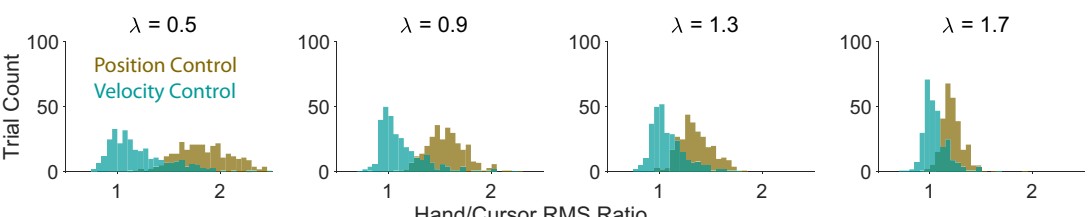

**Appendix 1—figure 3.** Distribution of hand/cursor RMS ratio over trials, calculated for simulated trials under Position Control (brown) and Velocity Control (cyan), for different $\lambda$ values.

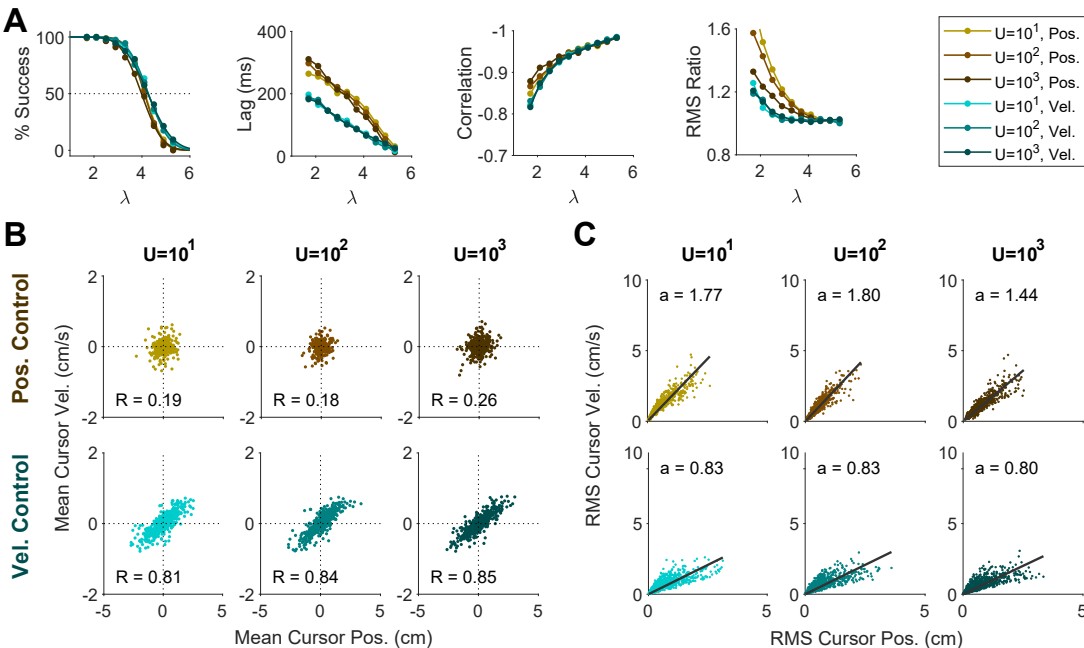

**Appendix 1—figure 4.** Effect of effort cost on control behavior represented by the (**A**) Aggregate performance measures, (**B**) Distribution of mean cursor movement, and (**C**) Distribution of RMS of cursor movement.

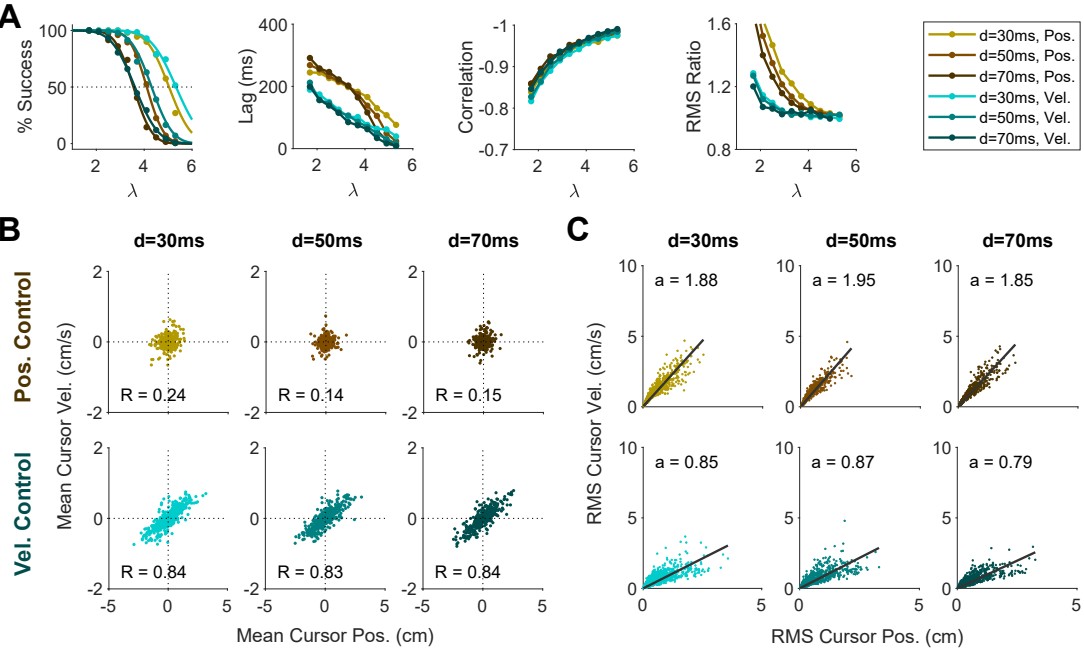

**Appendix 1—figure 5.** Effect of sensory delay on control behavior represented by the (**A**) Aggregate performance measures, (**B**) Distribution of mean cursor movement, and (**C**) Distribution of RMS of cursor movement.

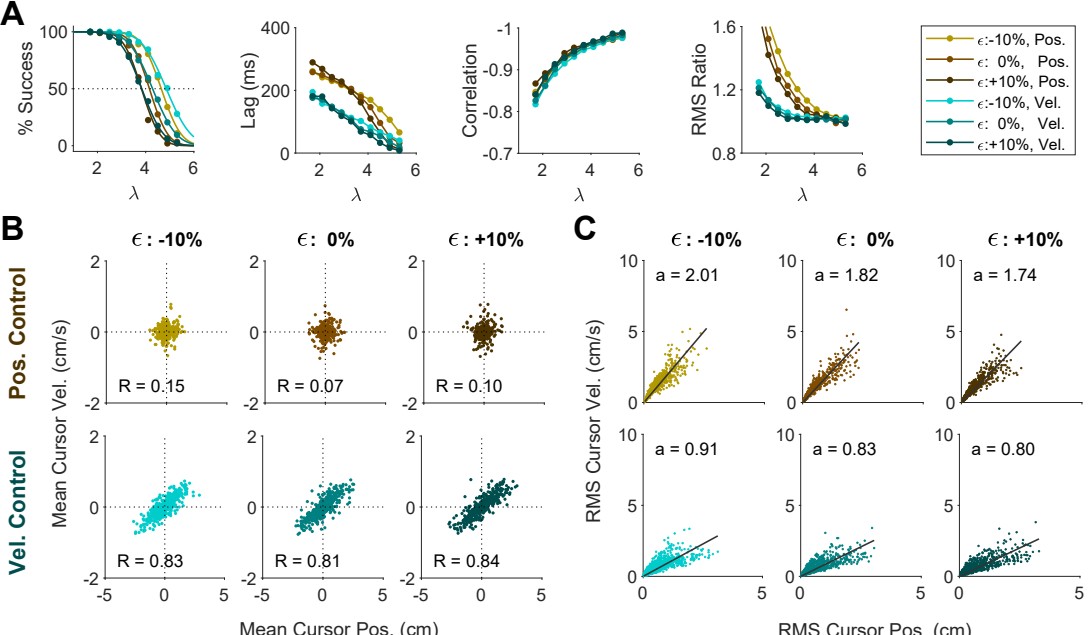

**Appendix 1—figure 6.** Effect of changing motor noise (10% reduction to 10% increase) on control behavior represented by the (**A**) Aggregate performance measures, (**B**) Distribution of mean cursor movement, and (**C**) Distribution of RMS of cursor movement.

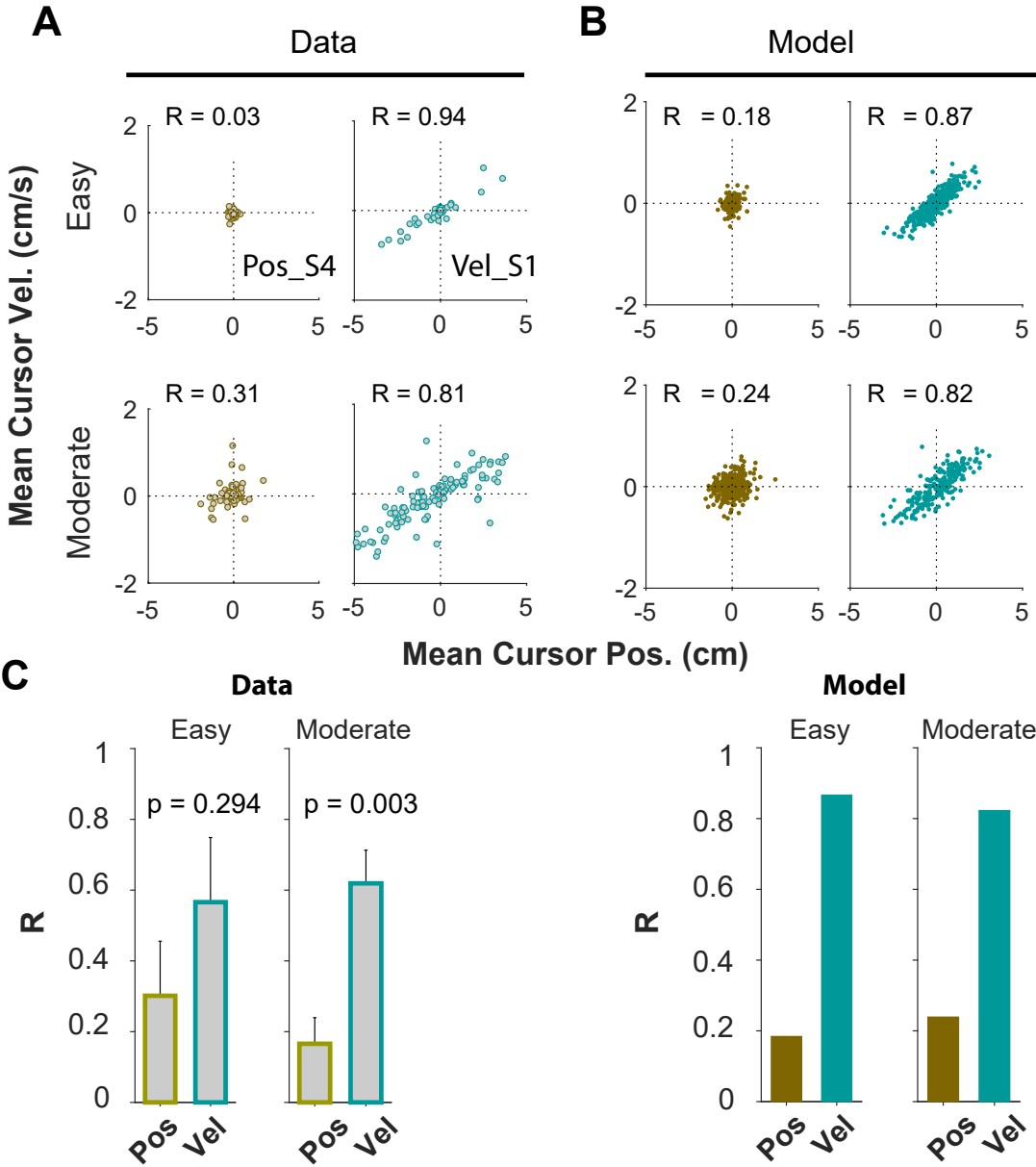

**Appendix 1—figure 7.** Joint distributions of cursor mean position and cursor mean velocity, separated into two difficulty levels: Easy ($\lambda \leq 70\% \lambda_c$), Moderate ($70\% \lambda_c < \lambda \leq \lambda_c$). (**A**) Experimental data and (**B**) Model data. (**C**) Correlation coefficient R between cursor mean position and mean velocity for different difficulty levels, plotted for data (left) and model (right). The error bars indicate standard error across subjects for each control objective group (n=6 per group).

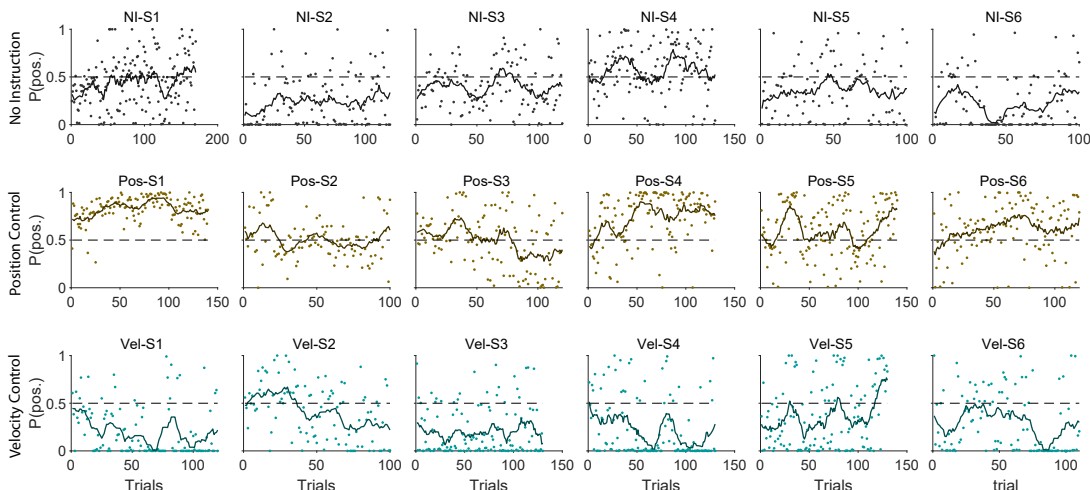

**Appendix 1—figure 8.** Probability of each trial performed under Position Control for each individual and group (top: Position Control group; bottom: Velocity Control group).

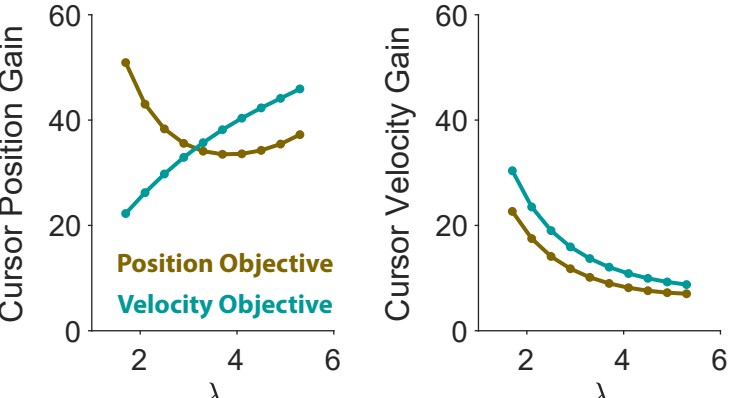

**Appendix 1—figure 9.** Optimal control gains corresponding to cursor position (left) and cursor velocity (right), obtained under Position Control (brown) and Velocity Control (cyan).

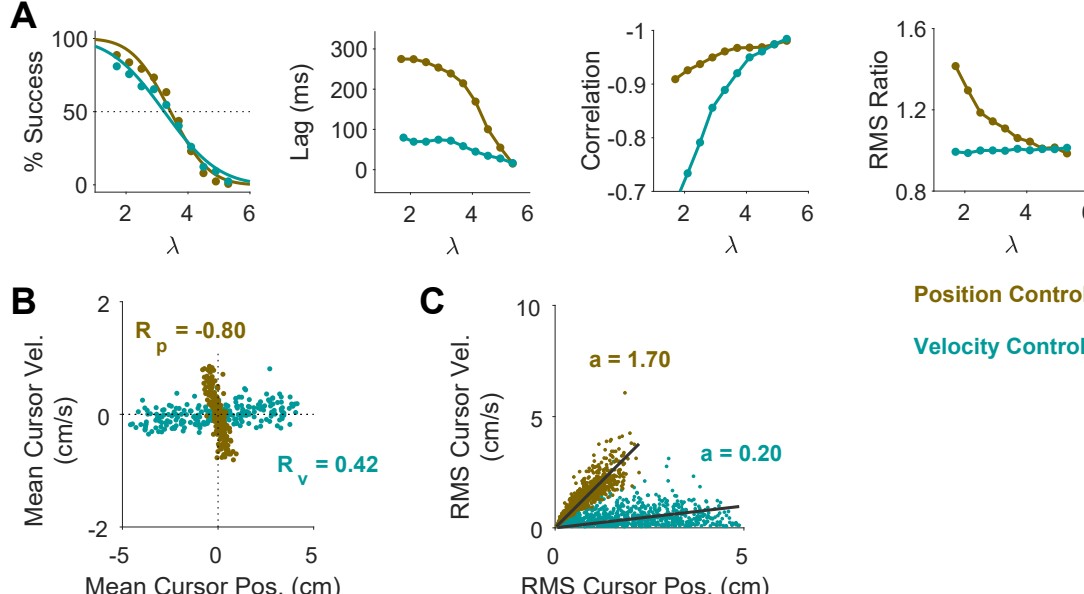

**Appendix 1—figure 10.** Model simulations of the CST task when introducing perturbations (random cursor jumps) at the start of each trial. (**A**) Aggregate performance of success rate, hand/cursor lag, correlation and RMS ratio as a function of difficulty level, shown for each control objective (brown: Position Control; cyan: Velocity Control). (**B**) Joint distribution of cursor mean position and cursor mean velocity under different control objectives. (**C**) Distribution of cursor RMS position and RMS velocity under different control objectives.

