## [Editor Report · eLife assessment]

This study represents a step towards integrating human and non-human primate research towards a broader understanding of the neural control of motor strategies. It could offer **valuable** insights into how humans and non-human primates (Rhesus monkeys) manage visuomotor tasks, such as stabilizing an unstable virtual system, potentially leading to discoveries in neural behaviour mechanisms. While the evidence is mostly **solid**, some results, particularly from the binary classification of control strategies for non instructed behaviour, require further validation before it could be conclusively interpreted.

---

## [Referee Report · Reviewer #1 (Public Review)]

The present study examines whether one can identify kinematic signatures of different motor strategies in both humans and non-human primates (NHP). The Critical Stability Task (CST) requires a participant to control a cursor with complex dynamics based on hand motion. The manuscript includes datasets on performance of NHPs collected from a previous study, as well as new data on humans performing the same task. Further human experiments and optimal control models highlight how different strategies lead to different patterns of hand motion. Finally, classifiers were developed to predict which strategy individuals were using on a given trial.

There are several strengths to this manuscript. I think the CST task provides a very useful behavioural task to explore the neural basis of voluntary control. While reaching is an important basic motor skill and commonly studied, there is much to learn by looking at other motor actions to address many fundamental issues on the neural basis of voluntary control.

I also think the comparison between human and NHP performance is important as there is a common concern that NHPs can be overtrained in performing motor tasks leading to differences in their performance as compared to humans. The present study highlights that there are clear similarities in motor strategies of humans and NHPs. While the results are promising, I would suggest that the actual use of these paradigms and techniques likely need some improvement/refinement. Notably, the threshold or technique to identify which strategy an individual is using on a given trial needs to be more stringent given the substantial overlap in hand kinematics between different strategies.

The most important goal of this study is to set up future studies to examine how changes in motor strategies impact neural processing. The revised manuscript has improved the technique for identifying which strategy appears to be performed by the individual. A pivotal assumption is that one can identify control strategies from differences in behaviour. As I'm sure the authors know, this inversion of the control problem is not trivial and so success requires that there are only a few 'reasonable' strategies to solve the control problem, and that these strategies lead to distinct patterns of behavior. Many of the concerns raised by myself and the other reviewers relate to this challenge. The revised manuscript now uses a more strict criteria which is good improvement.

One of the values of this paper is to start to develop the tools and approaches to address neural basis of control. The strength of the present manuscript is that it includes modelling, explicit strategy instructions in humans, and then analysis of free-form performance in humans and non-human primates. Given the novelty of this question and approach, there likely are many ways that the techniques and approaches could be improved, but I think they've done a great start. Their approach is quite clever and provides an important blueprint for future studies.

One weakness at this point is that there is still substantial overlap in behavoural performance predicted between strategies, as some human participants given an explicit strategy were almost equally categorized as reflecting the other strategy. I'm glad to see the addition of the model performance on perturbation trials as this additional figure clearly highlights much greater separation in performance than when observing natural behavior. While it is not reasonable to expand beyond this for the present manuscript, I think it is essential for this group to develop the perturbation paradigm (and potentially other approaches) that can better isolate behavioral signatures of different control strategies. I think future work will be strengthened by having multiple experimental angles to interpret the neural activity.

---

## [Referee Report · Reviewer #3 (Public Review)]

This paper considers a challenging motor control task - the critical stability task (CST) - that can be performed equally well by humans and macaque monkeys. This task is of considerable interest since it is rich enough to potentially yield important novel insights into the neural basis of behavior in more complex tasks that point-to-point reaching. Yet it is also simple enough to allow parallel investigation in humans and monkeys, and is also easily amenable to computational modeling. The paper makes a compelling argument for the importance of this type of parallel investigation and the suitability of the CST for doing so.

Behavior in monkeys and in human subjects suggests that behavior seems to include two qualitatively different kinds of behavior - in some cases, the cursor oscillates about the center of the screen, and in other cases, it drifts more slowly in one direction. The authors argue that these two behavioral regimes can be reliably induced by instructing human participants to either maintain the cursor in the center of the screen (position control objective), or keep the cursor still anywhere in the screen (velocity control objective) - as opposed to the usual 'instruction' to just not let the cursor leave the screen. A computational model based on optimal feedback control can reproduce the different behaviors under these two instructions.

Overall, this is a creative study that leverages experiments in humans and computational modeling to gain insight into the nature of individual differences in behavior across monkeys (and people). The authors convincingly demonstrate that they can infer the control objectives from participants who were instructed how to perform the task to emphasize either position or velocity control, based on the RMS cursor position and RMS cursor velocity. The authors show that, while other behavioral metrics do contain similar information about the control objective, RMS position and velocity are sufficient, and their approach classifies control objectives for simulated data with high accuracy (~95%).

The authors also convincingly show that the range of behaviors observed in the CST task cannot be explained as emerging from variations in effort cost, motor execution noise, or sensorimotor delays.

One significant issue, however relates to framing the range of possible control objectives as a simple dichotomy between 'position' and 'velocity' objectives. The authors do clearly state that this is a deliberate choice made in order to simplify their first attempts at solving this challenging problem. However, I do think that the paper at times gives a false impression that this dichotomous view of the control objectives was something that emerged from the data, rather than resulting from a choice to simplify the modeling/inference problem. For instance, line 115: "An optimal control model was used to simulate different control objectives, through which we identified two different control objectives in the experimental data of humans and monkeys."

In the no-instruction condition - which is the starting point and which the ultimate goal of the paper is to understand - there is a lot of variability in behavior across trials (even within an individual) and generally no clear correspondence to either the position or velocity objective. This variability is largely interpreted as the monkeys (and people) switching between control objectives on a trial-to-trial basis. If the behavior were truly a bimodal mixture of these two different behaviors, this might be a convincing interpretation. However, there are a lot of trials that fall in-between the patterns of behavior expected under the position and velocity control objectives. The authors do mention this issue in the discussion. However, it's not clearly examined whether these are simply fringe trials that are ambiguous (like some trials generated by the model are), or whether they reflect a substantial proportion of trials that require some other explanation (whether that is blended position/velocity control, or something else). The existence of these 'in-between' trials (which possibly amount to more than a third of all trials) makes the switching hypothesis a lot less plausible.

Overall, while I think the paper introduces a promising approach and overall helps to improve our understanding of the behavior in this task, I'm not fully convinced that the core issue of explaining the variability in behavior in the no-instruction condition (in monkeys especially) has been resolved. The main explanation put forward is that the monkeys are switching between control objectives on a trial-by-trial basis, but there is no real evidence in the data for this, and I don't think there is yet a good explanation of what is occurring in the 'in-between' trials that aren't explained well by velocity or position objectives.

---

## [Author Response]

The following is the authors’ response to the original reviews.

**Response to Public Reviewer Comments**

We again thank the reviewers for the time and effort they clearly put into reviewing our manuscript. We have revised our manuscript to take into account the majority of their suggestions, primary among them being refinements of our model and classification approach, detailed sensitivity analysis of our model, and several new simulations. Their very constructive feedback has resulted in what we feel is a much-improved paper. In what follows, we respond to each of their points.

**Reviewer #1:**
COMMENT: The reviewer suggested that our control policy classification thresholds should be increased, especially if the behavioral labels are to be subsequently used to guide analyses of neural data which “is messy enough, but having trials being incorrectly labeled will make it even messier when trying to quantify differences in neural processing between strategies.”

REPLY: We appreciate the observation and agree with the suggestion. In the revised manuscript, we simplified the model (as another reviewer suggested), which allowed for better training of the classifier. This enabled an increase in the threshold to 95% to have more confidence in the identified control strategies. Figures 7 and 8 were regenerated based on the new threshold.

COMMENT: The reviewer asked if we could discuss what one might expect to observe neurally under the different control policies, and also suggested that an extension of this work could be to explore perturbation trials, which might further distinguish between the two control policies.

REPLY: It is indeed interesting to speculate what neural activity could underlie these different behavioral signatures. As this task is novel to the field, it is difficult to predict what we might observe once we examine neural activity through the lens of these control regimes. We hope this will be the topic of future studies, and one aspect worthy of investigation is how neural activity prior to the start of the movement may reflect two different control objectives. Previous work has shown that motor cortex is highly active and specific as monkeys prepare for a cued movement and that this preparatory activity can take place without an imposed delay period (Ames et al., 2014; Cisek & Kalaska, 2005; Dekleva et al., 2018; Elsayed et al., 2016; Kaufman et al., 2014; Lara et al., 2018; Perich et al., 2018; Vyas et al., 2018; Zimnik & Churchland, 2021). It seems possible that the control strategies we observed correspond to different preparatory activity in the motor cortex. We added these speculations to the discussion.

The reviewer’s suggestion to introduce perturbations to probe sensory processing is very good and was also suggested by another reviewer. We therefore conducted additional simulations in which we introduced perturbations (Supplementary Material; Figure S10). Indeed, in these model simulations the two control objectives separated more. However, testing these predictions via experiments must await future work.

COMMENT: “It seems like a mix of lambda values are presented in Figure 5 and beyond. There needs to be some sort of analysis to verify that all strategies were equally used across lambda levels. Otherwise, apparent differences between control strategies may simply reflect changes in the difficulty of the task. It would also be useful to know if there were any trends across time?”

REPLY: We appreciate and agree with the reviewer’s suggestion. We have added a complementary analysis of control objectives with respect to task difficulty, presented in the Supplementary Material (Figures S7 and S8). We demonstrate that, overall, the control objectives remain generally consistent throughout trials and difficulty levels. Therefore, it can be concluded that the difference in behavior associated with different control objectives does not depend on the trial sequence or difficulty of the task. A statement to this extent was added to the main text.

COMMENT: “Figure 2 highlights key features of performance as a function of task difficulty. …However, there is a curious difference in hand/cursor Gain for Monkey J. Any insight as to the basis for this difference?”

REPLY: The apparently different behavior of Monkey J in the hand/cursor RMS ratio could be due to subject-to-subject variability. Given that we have data from only two monkey subjects, we examined inter-individual variations between human subjects in the Supplementary Material by presenting individual hand/cursor gain data for all individual human subjects (Figure S1). As can be seen, there was indeed variability, with some subjects not exhibiting the same clear trend with task difficulty. However, on average, the RMS ratio shows a slight decrease as trials grow more difficult, as was earlier shown in Figure 2. We added a sentence about the possibility of inter-individual variations to address the difference in behavior of monkey J with reference to the supplementary material.

**Reviewer #2:**

(Reviewer #2's original review is with the first version of the Reviewed Preprint. Below is the authors' summary of those comments.)

COMMENT: The reviewer commends the care and effort taken to characterize control policies that may be used to perform the CST, via dual human and monkey experiments and model simulations, noting the importance of doing so as a precursor to future neural recordings or BMI experiments. But the reviewer also wondered if it is all that surprising that different subjects might choose different strategies: “... it makes sense that different subjects might choose to favor different objectives, and also that they can do so when instructed. But has this taught us something about motor control or simply that there is a natural ambiguity built into the task?”

REPLY: The redundancy in the task that allowed different solutions to achieve the task was deliberate, and the motivation for choosing this task for this study. We therefore did not regard the resulting subject-to-subject variability as a finding of our study. Rather, redundancy and inter-individual variability are features ubiquitous in all everyday actions and we explicitly wanted to examine behavior that is closer to such behavior. As commended by the reviewers, CST is a rich task that extends our research beyond the conventional highly-constrained reaching task. The goal of our study was to develop a computational account to identify and classify such differences to better leverage future neural analyses of such more complex behaviors. This choice of task has now been better motivated in the Introduction of the revised manuscript.

COMMENT: The reviewer asks about our premise that subjects may use different control objectives in different trials, and whether instead a single policy may be a more parsimonious account for the different behavioral patterns in the data, given noise and instability in the system. In support of this view, the reviewer implemented a simple fixed controller and shared their own simulations to demonstrate its ability to generate different behavioral patterns simply by changing the gain of the controller. The reviewer concludes that our data “are potentially compatible with any of these interpretations, depending on which control-style model one prefers.”

REPLY: We first address the reviewer’s concern that a simple “fixed” controller can account for the two types of behavioral patterns observed in Experiment 2 (instructed groups) by a small change in the control gain. We note that our controller is also fixed in terms of the plant, the actuator, and the sensory feedback loop; the only change we explore is in the relative weights of position vs. velocity in the Q matrix. This determines whether it is deviations in position or in velocity that predominate in the cost function. This, in turn, generates changes in the gain vector L in our model, since the optimal solution (i.e. the gains L that minimize the cost function) depends on the Q matrix as well as the dynamics of the plant (specifically, the lambda value). Hence, one could interpret the differences arising from changes in the control objective (the Q matrix) as changes in the gains of our “fixed” controller.

More importantly, while the noise and instability in the system may indeed occasionally result in distinct behavioral patterns (and we have observed such cases in our simulations as well), these factors are far from giving an alternative account for the structural differences in the behavior that we attribute to the control objective. To substantiate this point, we performed additional simulations that are provided in the Supplementary Material (Figures S4—6). These simulations show that neither a change in noise nor in the relative cost of effort can account for the two distinct types of behavior. These differences are more consistently attributed to a change in the control objective.

In addition, our approach provides a normative account of the control gains needed to simulate the observed data, as well as the control objectives that underlie those gains. As such, the two control policies in our model (Position and Velocity Control) resulted in control gains that captured the differences in the experimental groups (Experiment 2), both at the single trial and aggregate levels and across different task difficulties. Figure S9 in the Supplementary Material shows how the control gains differ between Position and Velocity Control in our model across different difficulty levels.

We agree,with the reviewer’s overall point, that there are no doubt many models that can exhibit the variability observed in our experimental data, our simulations, or the reviewer’s simulations. Our study aimed to explore in detail not only the model’s ability to generate the variable behavior observed in experimental data, but also to match experimental results in terms of performance levels, gains, lags and correlations across a wide range of lambda values, wherein the only changes in the model were the lambda value and the control objective. Without the details of the reviewer’s model, we are unable to perform a detailed analysis of that model. Even so, we are not claiming that our model is the ‘ground truth,’ only that it is certainly a reasonable model, adopted from the literature, that provides intuitive and normative explanation about the performance of humans and monkeys over a range of metrics, system dynamics, and experimental conditions.

Finally, we understand the reviewer’s concern regarding whether the trial-by-trial identification of control strategy in Figure 8 suggests that (uninstructed) subjects constantly switch control objectives between Position and Velocity. Although it is not unreasonable to imagine that individuals would intuitively try different strategies between ‘keeping the cursor still’ and ‘keeping the cursor at the center’ across trials, we agree that it is generally difficult to determine such trial-to-trial changes, especially when the behavior lies somewhere in between the two control objectives. In such cases, as we originally discussed in the manuscript, an alternative explanation could be a mixed control objective that generates behavior at the intersection of Position and Velocity Control, i.e., between the two slopes in Figure 8. We believe, however, that our modeling approach is still helpful in cases where performance is predominantly based on Position or Velocity Control. After all, the motivation for this study was to parse neural data into two classes associated with each control objective to potentially better identify structure underlying these behaviors.

We clarified these points in the main text by adding further explanation in the Discussion section.

COMMENT: The reviewer suggested additional experiments, such as perturbation trials, that might be useful to further explore the separability of control objectives. They also suggested that we temper our conclusion that our approach can reliably discriminate amongst different control policies on individual trials. Finally, the reviewer suggested that we modify our Introduction and/or Discussion to note past human/monkey research as well as investigations of minimization of velocity-error versus position-error in the smooth pursuit system.

REPLY: We have expanded our simulations to investigate the effects of perturbation on the separability of different control objectives (Figure S10 in Supplementary Materials). We demonstrated that introducing perturbations more clearly differentiated between Position and Velocity Control. These results provide a good basis for further experimental verifications of the control objectives, but we defer these for future work.

We also appreciate the additional past work that bridges human and monkey research that the reviewer highlights, including the related discussions in the eye movement literature on position versus velocity control. We have modified our Introduction and Discussion accordingly.

**Reviewer #3:**
COMMENT: The reviewer asked whether the observed differences in behavior might be due to some other factors besides the control policy, such as motor noise or effort cost, and suggested that we more systematically ruled out that possibility.

REPLY: We appreciate and have heeded the reviewer’s suggestion. The revised manuscript now includes additional simulations in which the control objective was fixed to either Position or Velocity Control, while other parameters were systematically varied. Specifically, we examined the influence of the relative effort cost, the sensory delay, and motor noise, on performance. The results of these sensitivity analyses are presented in the Supplementary Material, Figures S4—6. In brief, we found that changing the relative effort cost, delay, or noise levels, mainly affected the success rate in performance (as expected), but did not affect the behavioral features originally associated with control objectives. We include a statement about this result in the main text with reference to the details provided in the Supplementary Material.

COMMENT: The reviewer questioned our choice of classification features (RMS position and velocity) and wondered if other features might yield better class separation, such as the hand/cursor gain. In a similar vein, reviewer 2 suggested in their recommendations that we examine the width of the autocorrelation function as a potentially better feature.

REPLY: We note first that our choice of cursor velocity and position stems from a dynamical systems perspective, where position-velocity phase-space analysis is common. However, we also explored other features as suggested. We found that they, too, exhibited overlap between the two different control objectives, and did not provide any significant improvement in classification performance (Figures S2 and S3; Supplementary Materials). Of course, that is not to say that a more exhaustive examination of features may not find ones that yield better classification performance than those we investigated, but that is beyond the scope of our study. We refer to this consideration of alternative metrics in the discussion.

COMMENT: The reviewer notes that “It seems that the classification problem cannot be solved perfectly, at least on a single-trial level.” To address this point, the reviewer suggested that we conduct additional simulations under the two different control objectives, and quantify the misclassifications.

REPLY: We appreciate the reviewer’s suggestion, and have conducted the additional simulations as suggested, the results of which are included in the revised manuscript.

COMMENT: “The problem of inferring the control objective is framed as a dichotomy between position control and velocity control. In reality, however, it may be a continuum of possible objectives, based on the relative cost for position and velocity. How would the problem differ if the cost function is framed as estimating a parameter, rather than as a classification problem?”

REPLY: A blended control strategy, formulated as a cost function that is a weighted combination of position and velocity costs, is indeed a possibility that we briefly discussed in the original manuscript. This possibility arises particularly for individuals whose performance metrics lie somewhere between the purely Position or purely Velocity Control. While our model allows for a weighted cost function, which we will explore in future work, we felt in this initial study that it was important to first identify the behavioral features unique to each control objective.

**Response to Recommendations for the Authors:**

**Reviewer #1 (Recommendations For The Authors):**

None beyond those stated above.

**Reviewer #2 (Recommendations For The Authors):**
COMMENT: Line 166 states "According to equation (1), this behavior was equivalent to reducing the sum (𝑝 + 𝑥) when 𝜆 increased, so as to prevent rapid changes in cursor velocity". This doesn't seem right. In equation 1, velocity (not acceleration) depends on p+x. So a large p+x doesn't create a "rapid change in cursor velocity", but rather a rapid change in cursor position.

REPLY: The reviewer is correct and we have corrected this misworded sentence; thank you for catching that.

COMMENT: The reviewer points out the potential confusion readers may have, given our unclear use of ‘control strategy’ vs. ‘control policy’ vs. ‘control objective’. The reviewer suggests that “It would be helpful if this could be spelled out early and explicitly. 'Control strategy' seems perilously close to 'control policy', and it would be good to avoid that confusion. The authors might prefer to use the term 'cost function', which is really what is meant. Or they might prefer 'control objective', a term that they introduce as synonymous with 'control strategy'.”

REPLY: We thank the reviewer for noting this ambiguity. We have clarified the language in the Introduction to explicitly note that by strategy, we mean the objective or cost function that subjects attempt to optimize. We then use ‘control objective’ consistently and removed the term ‘policy’ from the paper to avoid confusion. We also now use Position Control and Velocity Control as the labels for our two control objectives.

COMMENT: The reviewer notes that in Figure 2B and the accompanying text in the manuscript, we need to be clearer about what is being correlated; namely, cursor and hand position.

REPLY: Thank you for pointing out this lack of clarity, which we have corrected as suggested.

COMMENT: The reviewer questions our attribution of decreasing lag with task difficulty as a consequence of subjects becoming more attentive/responsive when the task is harder, and points out that our model doesn’t include this possible influence yet the model reproduces the change in lag. The reviewer suggests that a more likely cause is due to phase lead in velocity compared to position, with velocity likely increasing with task difficulty, resulting in a phase advance in the response.

REPLY: Our attribution of the decrease in lag with task difficulty being due to attention/motivation was a recapitulation of this point made in the paper by Quick et al. [2018]. But as noted by the reviewer, this potential influence on lag is not included in our model. Accordingly, the change in lag is more likely a reflection of the phase response of the closed loop system, which does change with task difficulty since the optimal gains depend upon the plant dynamics (i.e., the value of lambda). We have, therefore, deleted the text in question.

COMMENT: “The Methods tell us rather a lot about the dynamics of the actual system, and the cost functions are also well defined. However, how they got from the cost function to the controller is not described. I was also a bit confused about the controller itself. Is the 50 ms delay assumed when deriving the controller or only when simulating it (the text seems to imply the latter, which might make sense given that it is hard to derive optimal controllers with a hard delay)? How similar (or dissimilar) are the controllers for the two objectives? Is the control policy (the matrix that multiplies state to get u) quite different, or only subtly?”

REPLY: Thanks for pointing this out. For brevity, we had omitted the details and referred readers to the original paper (Todorov, 2005). However, we now revised the manuscript to now include all the details in the Methods section. Hence, the entire section on the model is new. This also necessitated updating all data figures (Figures 3, 4, 5, 6, 7, 8) as they contain modeling results.

COMMENT: “Along similar lines, I had some minor to moderate confusions regarding the OFC model as described in the main text. Fig 3 shows a model with a state estimator, but it isn't explained how this works. …Here it isn't clear whether there is sensory noise, or a delay. The methods say a delay was included in the simulation (but perhaps not when deriving the controller?). Noise appears to have been added to u, but I'm guessing not to x or x'? The figure legend indicates that sensory feedback contains only some state variables, and that state estimation is used to estimate the rest. Presumably this uses a Kalman filter? Does it also use efference copy, as would be typical? My apologies if this was stated somewhere and I missed it. Either way, it would be good to add a bit more detail to the figure and/or figure legend.”

REPLY: As the lack of detail evidently led to some confusion, we now more clearly spell out the details of the model in the Methods, including the state estimation procedure.

COMMENT: The reviewer wondered why we chose to plot mean velocity vs. mean position as in Figure 5, noting that, “ignoring scale, all scatter plots would be identical if the vertical axis were final position (because mean velocity determines final position). So what this plot is really examining is the correlation between final position and average position. Under position control, the autocorrelation of position is short, and thus final position tends to have little to do with average position. Under velocity control, the autocorrelation of position is long, and thus final position tends to agree with average position. Given this, why not just analyze this in terms of the autocorrelation of position? This is expected to be much broader under velocity control (where they are not corrected) than under position control (where they are, and thus disappear or reverse quickly). To me, thinking of the result in terms of autocorrelation is more natural.”

REPLY: The reviewer is correct that the scatter plots in Fig. 5 would be the same (to within a scale factor of the vertical axis) had we plotted final position vs. mean position instead of mean velocity vs. mean position as we did. Our preference for mean velocity vs. mean position stems from a dynamical systems perspective, where position-velocity phase-space analysis is common. We now mention these perspectives in the revised manuscript for the benefit of the reader.

As suggested, we also investigated the width of the (temporal) autocorrelation function (acf) of cursor position for 200 simulated position control trials and 200 simulated velocity control trials, at four different lambda values (50 simulated trials per lambda). Figs. S2A and B (Supplementary Materials) show example trials and histograms of the acf width, respectively. As the reviewer surmised, velocity control trials tend to have wider acfs than position control trials. However, as with the metrics we chose to analyze, there is overlap and there is no visible benefit for the classification.

COMMENT: “I think equation ten is incorrect, but would be correct if the identity matrix were added? Also, why is the last term of B set to 1/(Tau*M). What is M? Is it mass (which above was lowercase m)? If so, mass should also be included in A (it would be needed in two places in the last column). Or if we assume m = 1, then just ignore mass everywhere, including here and equation 5. Or perhaps I'm confused, and M is something else?”

REPLY: Thanks for pointing this out. The Matrix A shown in the paper is for the continuous-time representation of the model. However, as the reviewer correctly mentioned, for the discrete-time implementation of the model, a modification (identity matrix) was added in our simulations. We have now clarified this in the Methods section of the revised manuscript. Also, as correctly pointed out, M is the mass of the hand, which depending on whether the hand acceleration (d^2 p/dt^2) or hand force (F) are taken as the state, it can be included in the A matrix. In our case, the A matrix is modified according to the state vector. Similarly, the B matrix is also modified. This is now clarified in the Methods section of the manuscript.

**Reviewer #3 (Recommendations For The Authors):**
COMMENT: “Equations 4-8 are written in continuous time, but Equation 9 is written in discrete time. Then Equation 10 is in discrete time. This needs to be tidied up. … I would suggest being more detailed and systematic, perhaps formulating the control problem in continuous time and then converting to discrete time.”

REPLY: Thank you for this helpful suggestion. The model section in the Methods has been expanded to provide further details of the equation of motion, the discretization process, the control law calculation and the state estimation process.

COMMENT: “It seems slightly odd for the observation to include only position and velocity of the cursor. Presumably participants can also observe the state of their own hand through proprioception (even if it were occluded). How would it affect the model predictions if the other states were observable?”

REPLY: Thanks for pointing this out. We initially included only cursor position and velocity since we felt that was the most prominent state feedback, and the system is observable in that case. Nevertheless, we revised the manuscript and repeated all simulations using a full observability matrix. Our findings and conclusions remain unchanged. With the changes in the modeling, the figures were also updated (Fig.3, 4, 5, 6, 7, 8).

COMMENT: “It seems unnecessary to include the acceleration of the cursor in the formulation of the model. …the acceleration is not even part of the observed state according to line 668… I think the model could therefore be simplified by omitting cursor acceleration from the state vector.”

REPLY: We agree. We have simplified the model, and generated new simulations and figures. Our results and conclusions were unchanged by this modification. With the changes in the modeling, the figures were also updated (Fig.3, 4, 5, 6, 7, 8).

COMMENT: “In the cost function, it's not clear why any states other than position and velocity of the cursor need to have non-zero values. …The choice to have the cost coefficient for these other states be 1 is completely arbitrary… If the point is that the contribution of these other costs should be negligible, then why not just set them to 0?”

REPLY: We agree, and have made this change in the Methods section. Our findings and conclusions were unaffected.

COMMENT: “It seems that the cost matrices were specified after transforming to discrete-time. It is possible however (and perhaps recommended) to formulate in continuous time and convert to discrete time. This can be done cleanly and quite straightforwardly using matrix exponentials. Depending on the discretization timestep, this can also naturally lead to non-zero costs for other states in the discrete-time formulation even if they were zero under continuous time. … A similar comment applies to discretization of the noise.”

REPLY: Thanks for the suggestion. We have expanded on the discretization process in our Methods section, which uses a common approximation of the matrix exponentiation method.

COMMENT: “Most of the parameters of the model seem to be chosen arbitrarily. I think this is okay as the point is to illustrate that the kinds of behaviors observed are within the scope of the model. However, it would be helpful to provide some rationale as to how the parameters were chosen. e.g. Were they taken directly from prior literature, or were they hand-tuned to approximately match observed behavior?”

REPLY: We have revised the manuscript to more clearly note that the noise parameters, as well as parameters of the mechanical system (mass, muscle force, time scale, etc) in our model were taken from previous publications (Todorov, 2005, Cluff et al. 2019). As described in the manuscript, the parameter values of the cost function (Q matrix) were obtained by tuning the parameters to achieve a similar range of success rate with the model as observed in the experimental data. This is now clarified in the Methods section.

COMMENT: “The ‘true’ cost function for this task is actually a 'well' in position space - zero cost within the screen and very high cost elsewhere. In principle, it might be possible to derive the optimal control policy for this more veridical cost function. It would be interesting to consider whether or not this model might reproduce the observed behaviors.”

REPLY: This is indeed a very interesting suggestion, but difficult to implement based on the current optimal feedback control framework. However, this is interesting to consider in future work.

Minor Comments:COMMENT: “In Figs 4 and 5, the data points are drawn from different conditions with varying values of lambda. How did the structure of this data depend on lambda? Might it be possible to illustrate in the figure (e.g. the shade/color of each dot) what the difficulty was for each trial?”

REPLY: We performed additional analyses to show the effects of task difficulty on the choice of control objective. Overall, we found that the main behavioral characteristics of the control objective remained fairly unchanged across different task difficulties or across time. The results of this analysis are included in Fig. S7 and S8 of the Supplementary Materials.

COMMENT: “Should mention trial duration (6s) in the main narrative of the intro/results.”

REPLY: We now mention this detail when we describe the task for the first time.

COMMENT: “As an alternative to training on synthetic data (which might not match behavior that precisely, and was also presumably fitted to subject data at some level) it might be worth considering to do a cross-validation analysis, i.e. train the classifier on subsets of the data with one participant removed each time, and classify on the held-out participant.”

REPLY: This is indeed a valid point. The main reason to train the classifier based on model simulations was two-fold: first, to have confidence in the training data, as the experimental data was limited and noisy, which would result in less reliable classifications; and second, the model simulations are available for different contexts and conditions, where experimental data is not necessarily available. The latter is a more practical reason to be able to identify control objectives for any subject (who received no instructions), without having to collect training data from matching control subjects who received explicit instructions. Nonetheless, we appreciate the reviewer’s recommendation and will consider that for our future studies.

COMMENT: “line 690 - Presumably the optimal policy was calculated without factoring in any delay (this would be tricky to do), but the 50ms delay was incorporated at the time of simulation?”

REPLY: The discretization of the system equations allowed us to incorporate the delay in the system dynamics and solve for the optimal controller with the delay present. This was done simply by system augmentation (e.g., Crevecoeur et al., 2019), where the states of the system in the current time-step were augmented with the states from the 5 preceding time-steps to form the new state vector x(t)_aug = [x(t) , x(t-1) , … , x(t-d) ]. Similarly, the matrices A, B, and H from the system dynamics could be expanded accordingly to form the new dynamical system:

$$x(t+1)*{aug} = A*{aug} * x(t)*{aug} + B*{aug} * u$$

Then, the optimal control was implemented on the new (augmented) system dynamics.

We have revised the manuscript (Methods) to clarify this issue.